# Identifying Selections
# for Unsupervised Subtask Discovery

**Yiwen Qiu**
Carnegie Mellon University
Pittsburgh, PA 15213
yiwenq@andrew.cmu.edu

**Yujia Zheng**
Carnegie Mellon University
Pittsburgh, PA 15213
yujiazh2@andrew.cmu.edu

**Kun Zhang**[*]
Carnegie Mellon University, MBZUAI
Pittsburgh, PA 15213
kunz1@andrew.cmu.edu

## Abstract

When solving long-horizon tasks, it is intriguing to decompose the high-level task into subtasks. Decomposing experiences into reusable subtasks can improve data efficiency, accelerate policy generalization, and in general provide promising solutions to multi-task reinforcement learning and imitation learning problems. However, the concept of subtasks is not sufficiently understood and modeled yet, and existing works often overlook the true structure of the data generation process: subtasks are the results of a *selection* mechanism on actions, rather than possible underlying confounders or intermediates. Specifically, we provide a theory to identify, and experiments to verify the existence of selection variables in such data. These selections serve as subgoals that indicate subtasks and guide policy. In light of this idea, we develop a sequential non-negative matrix factorization (seq-NMF) method to learn these subgoals and extract meaningful behavior patterns as subtasks. Our empirical results on a challenging Kitchen environment demonstrate that the learned subtasks effectively enhance the generalization to new tasks in multi-task imitation learning scenarios. The codes are provided at this [link].

## 1   Introduction

Being able to reuse learned skills from past experiences and conduct hierarchical planning (LeCun [2022], Hafner et al. [2022], Gehring et al. [2021]) is crucial for tackling real-world challenging tasks such as driving: it is meaningful to let higher levels of abstractions perform longer-term prediction (of subgoals), while lower levels of policy perform shorter-term actions. The concept of breaking down a task into subtasks[1] can be illustrated through the example of commuting to New York. You are at home, and the overall task, commuting to New York, can be decomposed into smaller, manageable subtasks. This includes subtasks like walking out of the house, getting into the car, driving and catching an airplane. Even more granular, each of these subtasks can be further broken down into smaller actions: walking out of the house involves standing up, grabbing the luggage and walking to the door. This method of decomposing a task into subtasks fits our intuition of how humans perform actions and helps to simplify complex tasks, making them more manageable.

---

[*]Corresponding author.

[1]We predominantly use the concept of *subtask* in this paper, which has similar semantics as that of *option* or *skill* in other literature.

38th Conference on Neural Information Processing Systems (NeurIPS 2024).

For artificial intelligence (AI) to match the ability of humans in terms of understanding the event structures (Zacks et al. [2001], Baldassano et al. [2017]) and learning to perform complex and long-horizon tasks by reinforcement learning (RL) (Sutton and Barto [2018]) or imitation learning (IL) (Hussein et al. [2017]), it is natural to ask this question: with an abundance of past collected either human or robotics experiences, how can one extract reusable subtasks, such that we can use them to solve future unseen new complex tasks? Current RL is well known for its sample inefficiency, and learning decomposition of subtasks serves as the basis for perform complex tasks via planning. The benefit is straightforward: extracting reusable disentangled (Denil et al. [2017]) temporal-extended common structure can enhance data efficiency and accelerate learning of new tasks (Thrun and O'Sullivan [1996], Florensa et al. [2017], Griffiths et al. [2019], Jing et al. [2021]). This is the main motivation for subtask discovery, the problem we aim to thoroughly investigate in this paper.

In subtask discovery, the criterion to segment these subtasks is vital, and should be consistent with the true generating process. However, most prior works did not explore the ground-truth generative structure to identify that criterion–instead, they simply focus on designing heuristic algorithms (McGovern and Barto [2001], Şimşek and Barto [2004], Wang et al. [2014], Paul et al. [2019]) or maximizing the likelihood of the data sequences that is built upon intuitive graphical models (Krishnan et al. [2017], Kipf et al. [2019], Sharma et al. [2019b], Yu et al. [2019]). Relying on a structure that conflicts with the true generating process may mislead the segmentation as well as worsen the performance of those downstream IL tasks. Thus, we argue that it is helpful to consider the causal structure underlying the sequence, specifically, the *selection* variables as subgoals to discover subtasks.

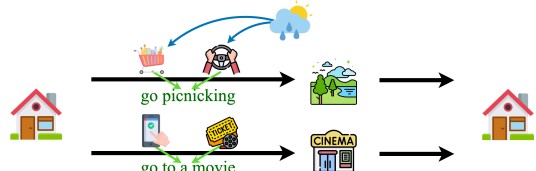

Figure 1: Example of subgoals as selections. One subgoal is to "go picnicking", another subgoal is to "go to a movie". In order to "go picnicking", you need to go shopping first and then drive to the park; in order to "go to a movie", you need to check the movie information online first and then get the tickets. The actions *caused* us to accomplish the subtasks, and we essentially select the actions based on (conditioned on) the subgoals we want to achieve. On the contrary, weather is a confounder of the states and actions: changing our actions would not influence the weather, but actions influence whether we can achieve the subgoals.

Selective inclusion of data points is prevalent in practical scenarios. We provide a short illustration of selection in Fig.1 to help distinguish the confounder and the selection case. We are at home, and have two subgoals to select from: "picnicking" and "movie", and go back home again. One way to look at selection is recognizing its preferential nature of including certain data (Heckman [1979]): achieving a subgoal involves performing a subtask, which consists of a sequence of states and actions that must follow a unique pattern. Another way to look at selection is by considering interventions (Eberhardt and Scheines [2007]) which is a widely used concept in the literature on causal discovery. Intervention involves assigning values to a single variable: in our case, the actions lead us to achieve a subgoal, and intervening on the actions would lead us to achieve others. In other words, the subgoal functions as a *selection* for the actions. Overlooking the selection structure is adopting an inappropriate inductive bias, and it can distort our understanding of the data. Therefore, the incorporation of identifying and modeling the selection structure to uncover the true causal process is crucial for understanding subtasks. A more comprehensive literature review on recent advances in understanding selections and subtask discovery can be found in Appx. A.

The main contributions of this paper are as follows: (1) We show that one can identify the selection structure without interventional experiments, and confirm its existence through our experiments. (2) According to the identified structure, and based on the formal definition of subtasks, we propose a novel sequential non-negative matrix factorization (seq-NMF) method to learn subgoals as selections. (3) We demonstrate that the learned subtasks can be leveraged to improve the performance of imitation learning in solving new tasks.

## 2   Preliminary

**Imitation Learning.**   In standard Imitation Learning, we collected a set of trajectories $\mathcal{D} = \{\tau_n\}_{n=1}^{N}$ from an expert in a Markov Decision Process (MDP). An MDP is defined by $\langle \mathcal{S}, \mathcal{A}, \mathcal{P}, \mathcal{R}, \rho_0, \gamma \rangle$, with $\mathcal{S}$ as the state space, $\mathcal{A}$ as the action space, $\mathcal{P} : \mathcal{S} \times \mathcal{A} \times \mathcal{S} \to [0, 1]$ as the transition probability, $\mathcal{R} : \mathcal{S} \times \mathcal{A} \to \mathbb{R}$ as the reward function, $\rho_0$ as the initial state distribution, and $\gamma$ as the discount factor.

Suppose we are given $N$ expert trajectories $\tau_n$ generated in an MDP with unknown reward. Each trajectory $\tau_n = \{s_t, a_t, \cdots\}_{t=1}^{T^n}$ is composed of a series of states $s_t \in \mathcal{S}$ and actions $a_t \in \mathcal{A}$. The goal of the agent is to learn a policy $\pi(a_t \mid s_t) : \mathcal{S} \times \mathcal{A} \to [0, 1]$ that mimics the expert's behavior.

**Option Framework.** For multi-task settings, learning a hierarchical policy that extracts basic skills has been proven useful in solving tasks composed of multiple subtasks. Sutton et al. [1999] proposed the option framework, which is a hierarchical reinforcement learning framework that decomposes a task into a set of temporally extended options. An option $\mathcal{O}$ is defined as a tuple $\langle \mathcal{I}, \pi, \beta \rangle$, where $\mathcal{I} \subseteq \mathcal{S}$ is the initiation set, $\pi$ is the policy, and $\beta : \mathcal{S} \to [0, 1]$ is the termination function deciding whether the current option terminates. The execution of a policy within an option is a semi-Markov decision process (SMDP). By sequentially performing SMDPs, that is, taking an option $\mathcal{O}_a$ until it terminates, and then taking the next option $\mathcal{O}_b$ until it terminates, the agent can learn a hierarchical policy that executes in the overall MDP.

**Causal Graph and Common Assumptions.** In a Bayesian network, the distribution $\mathbb{P}$ over a set of variables is assumed to be *Markov* w.r.t. to a directed acyclic graph (DAG) $\mathcal{G} = \{\mathcal{V}, \mathcal{E}\}$ where $\mathcal{V}$ is the set of vertices and $\mathcal{E}$ is the set of edges, and the DAG $\mathcal{G}$ is *faithful* to the data. The Markov condition, and faithfulness assumption are defined in Def. 1 and Def. 2, respectively.

**Definition 1.** (Markov Condition (Spirtes et al. [2001], Pearl [2009])) Given a DAG $\mathcal{G}$ and distribution $\mathbb{P}$ over the variable set $\mathcal{V}$, every variable X in $\mathcal{V}$ is probabilistically independent of its non-descendants given its parents in $\mathcal{G}$.

**Definition 2.** (Faithfulness Assumption (Spirtes et al. [2001], Pearl [2009])) There are no independencies between variables that are not entailed by the Markov Condition.

Combining the Markov Condition and Faithfulness Assumption, we can use *d-separation* as a criterion (denoted as $\mathbf{X} \perp_d \mathbf{Y} \mid \mathbf{Z}$) to read all the conditional independencies from a given DAG $\mathcal{G}$:

**Definition 3.** (d-separation (Spirtes et al. [2001], Pearl [2009])) Two sets of nodes $\mathbf{X}$ and $\mathbf{Y}$ in $\mathcal{G}$ is said to be *d-separated* by a set of nodes $\mathbf{Z} \subseteq \mathcal{V}$ if and only if: for every path $p$ that connects one node $i$ in $\mathbf{X}$ to one node $j$ in $\mathbf{Y}$, either (1) $p$ contains $i \to m \to j$ or $i \leftarrow m \to j$ such that $m$ is in $\mathbf{Z}$, or (2) $p$ contains a collider $m$, i.e. $i \to m \leftarrow j$ such that $m$ and all descendants of $m$ are not in $\mathbf{Z}$.

**Problem Formulation** Given the above context, we formulate the considered imitation learning problem as follows: we have a distribution of tasks $\mathcal{P}_e(\mathcal{T})$ and a corresponding set of expert trajectories $\mathcal{D} = \{\tau_n\}_{n=1}^N$, and we aim to let the imitater learn a policy that can be transferred to new tasks that follow a different distribution $\mathcal{P}_i(\mathcal{T})$. Each task sampled from $\mathcal{P}.(\mathcal{T})$ should be generated by a MDP and composed of a sequence of option $\{\mathcal{O}_j, \cdots\}$, where $\mathcal{O}_j = \langle \mathcal{I}_j, \pi_j, \beta_j \rangle_j$. We use $\mathcal{O} = \bigcup_{j=1}^J \mathcal{O}_j$ to denote all J options, and $\xi_p = \{\mathbf{s_t}, \mathbf{a_t}, ...\}_{t=1}^{\leq L}$ as a sub-sequence of states and actions $(\mathbf{s_t}, \mathbf{a_t})$ from any trajectory $\tau_n$. Each trajectory can be partitioned into sub-sequences $\xi\_p$ with maximum length L. Unlike traditional MIL (Seyed Ghasemipour et al. [2019], Yu et al. [2019]), we assume a shift in the task distribution, i.e. $\mathcal{P}_i(\mathcal{T}) \neq \mathcal{P}_e(\mathcal{T})$, but they only share the same option set, and we expect the agent to leverage the past learned subtasks to carry out new tasks efficiently.

**Roadmap for solving the "subtask discovery" problem** The question raised above on how to discover useful subtasks for policy generalization on new tasks can be answered two-fold. First, it is essential to adopt an accurate understanding of subtasks that aligns with a true data generation process: it is a matter of comprehending temporal data dynamics (Matsubara et al. [2014], Chen et al. [2018], Dyn), to which we provide an answer in Sec. 3.1. Second, based on that understanding, we need to design a learning algorithm that can extract subtasks from expert demonstrations, as discussed in Sec. 3.2. Finally, the learned subtasks should be used to facilitate policy training so that the policy can quickly adapt to new tasks by alternating between subtasks, which we address in Sec. 4.

# 3 Subgoal as Selection

**Overview** The first step is to understand *what* a subtask is. We propose to understand subtask by building the causal graph and distinguishing different potential structures. We assert that a subtask is indicated by a selection variable denoted as $\mathbf{g_t}$ (subgoal), and will distinguish it from confounder and intermediate node (Sec. 3.1). Then, we give a formal definition of subtasks as sub-sequences that can be representative of common behavior patterns and avoid uncertainties in policy (Sec. 3.2). With the understanding of selection (Sec. 3.1) and formal definition (Sec. 3.2), we propose a novel sequential non-negative matrix factorization (seq-NMF) method that aligns with these two ideas to learn subtasks from multi-task expert demonstrations (Sec. 3.3).

## 3.1 Identifying Selections in Data

To better understand subtasks, we build a DAG $\mathcal{G} = \{\mathcal{V}, \mathcal{E}\}$ to represent the generation process of a trajectory by setting each $\mathbf{s_t}, \mathbf{a_t}$ as vertices ($\mathcal{V}$), and add edges ($\mathcal{E}$) by connecting $\mathbf{s_t} \to \mathbf{s_{t+1}}, \mathbf{a_t} \to \mathbf{s_{t+1}}$ to represent the transition function $\mathcal{P}$. There are three potential patterns as follows:

**Definition 4.** (Confounder, selection and intermediate node) $\mathbf{c_t}$ is a *confounder* if $\mathbf{s_t} \leftarrow \mathbf{c_t} \to \mathbf{a_t}$, $\mathbf{g_t}$ is a *selection* if $\mathbf{s_t} \to \mathbf{g_t} \leftarrow \mathbf{a_t}$, $\mathbf{m_t}$ is an *intermediate node* if $\mathbf{s_t} \to \mathbf{m_t} \to \mathbf{a_t}$.

In other words, the causal Dynamic Bayesian Network (DBN) (Murphy [2002]) of a series of $\{\mathbf{s_t}, \mathbf{a_t}, \cdots\}$ is one of the following three scenarios:

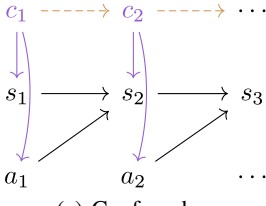
(a) Confounders

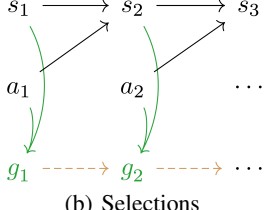
(b) Selections

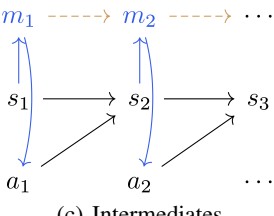
(c) Intermediates

Figure 2: Three kinds of dependency patterns of DAGs that we aim to distinguish. Structure (1) models the confounder case $\mathbf{s_t} \leftarrow \mathbf{c_t} \to \mathbf{a_t}$, structure (2) models the selection case $\mathbf{s_t} \to \mathbf{g_t} \leftarrow \mathbf{a_t}$, and structure (3) models the mediator case $\mathbf{s_t} \to \mathbf{m_t} \to \mathbf{a_t}$. In all three scenarios, the solid black arrows ($\to$) indicate the transition function that is invariant across different tasks. The dashed arrows ($\to$) indicate dependencies between nodes $\mathbf{d_t}$ and $\mathbf{d_{t+1}}$. We take them to be direct adjacencies in the main paper, and for potentially higher-order dependencies, we refer to Appx. B.4.

Selection implies that we can only observe the data points for which the selection criterion is met, such as reaching some subgoal $j$, e.g. $\mathbf{g_t}^{(j)} = 1$. As a consequence, the trajectory distribution $p(\mathbf{s_t}, \mathbf{a_t})$ is actually the conditional one: $p(\mathbf{s_t}, \mathbf{a_t} \mid \mathbf{g_t})$. Understanding subgoals as selections that are always *conditioned on* enables us to explain the relationship between the states and actions: the dependency between $\mathbf{s_t}$ and $\mathbf{a_t}$ is the result of the given selection, i.e. $\mathbf{s_t} \not\perp\!\!\!\perp \mathbf{a_t} \mid \mathbf{g_t}$. We argue that the policy $\pi(\mathbf{a_t} \mid \mathbf{s_t})$ does not indicate a direct causal relation between states and action, but rather an inference from states to actions, and the dependency is built by subgoal as selections.

We start by proposing a conditional independence test (CI test) based condition in Prop. 1 as a *sufficient* condition for recognizing selections:

**Proposition 1.** (Sufficient condition) Assuming that the graphical representation is Markov and faithful to the measured data, if $\mathbf{s_t} \not\perp\!\!\!\perp \mathbf{a_t} \mid \mathbf{d_t}$, then $\mathbf{d_t}$ is a selection variable, i.e., $\mathbf{d_t} := \mathbf{g_t}$, under the assumption that:

1. (confounder, selection, and intermediate nodes can not co-exist) At each time step, $\mathbf{d_t}$ can only be one of $\{\mathbf{c_t}, \mathbf{g_t}, \mathbf{m_t}\}$. (For a relaxation of this assumption, see Appx. B.4).

2. (consistency in a time series) At every time step, $\mathbf{d_t}$ plays the same role as one of $\{\mathbf{c_t}, \mathbf{g_t}, \mathbf{m_t}\}$.

For a *necessary and sufficient* condition, we have the following proposition:

**Proposition 2.** (Necessary and sufficient condition) $\mathbf{d_t}$ is a selection variable ($\mathbf{d_t} := \mathbf{g_t}$) if and only if $\mathbf{s_t} \not\perp\!\!\!\perp \mathbf{a_t} \mid \mathbf{d_t}$ and $\mathbf{d_t} \not\perp\!\!\!\perp \mathbf{a_{t+1}} \mid \mathbf{d_{t+1}}$.

Since our goal is to identify the presence of selection, the necessity aspect is not our primary focus, but it is still worthwhile to consider. For a condition that is weaker but has real-world implications, we have the following *necessary* condition:

**Proposition 3.** (Necessary condition) If $\mathbf{d_t}$ is a selection variable ($\mathbf{d_t} := \mathbf{g_t}$), then $s_{t+1} \perp\!\!\!\perp g_t \mid s_t, a_t$. (Such independency does not hold true for confounders case which is discussed in Appx. B.3)

This proposition implies that $s_{t+1}$ is only determined by $s_t, a_t$ and solely relies on the transition function in the environment. The hidden variable $g_t$ would not provide much additional information for the prediction of $s_{t+1}$. On the other hand, if we consider the confounder case, it is unrealistic that changing the subtask would result in changing the transition function, as the confounder case entails.

Such is strong evidence for us to assert that subtasks should be considered as selections. The proof of Prop. 1, Prop. 2 and Prop. 3 are provided in Appx. B.1, B.2, B.3, respectively.

**Remark 1.** As a relaxation of Prop. 1, we do not assert $\mathbf{c_t}$, $\mathbf{g_t}$ and $\mathbf{m_t}$ to be mutually exclusive. As long as we have $\mathbf{s_t} \not\perp\!\!\!\perp \mathbf{a_t} \mid \mathbf{d_t}$ and $\mathbf{d_t} \not\perp\!\!\!\perp \mathbf{a_{t+1}} \mid \mathbf{d_{t+1}}$, then the selection mechanism holds. There can also be confounders and intermediate nodes, but for simplicity, we do not consider them in the main paper and leave the combination of multiple $\mathbf{c_t}$, $\mathbf{g_t}$ and $\mathbf{m_t}$ to Appx. B.4.

**Remark 2.** There is a parallel between the selection variable and the reward signal in reinforcement learning from a probabilistic inference view (Levine [2018]). See discussion in Appx. E.

Based on Prop. 2, we verify that there is indeed selection in the trajectories, as indicated by experiments (Sec. 5.1). These selection variables serve as subgoals that can facilitate imitation learning.

## 3.2 Definition of the Subtask

In Prop. 1, we provide the sufficient condition to identify selection (subgoal), that is $\mathbf{s_t} \not\perp\!\!\!\perp \mathbf{a_t} \mid \mathbf{g_t}$. In other words, the current action $\mathbf{a_t}$ is affected by both the current state $\mathbf{s_t}$ and the current subgoal $\mathbf{g_t}$. Only $\mathbf{s_t}$ alone is not sufficient to determine action, but also the current subgoal $\mathbf{g_t}$ guides the agent's action. e.g. When arriving at a crossroad, and you are deciding whether to turn left or right, it is only when a subgoal (the left road or the right road) is selected, then the subgoal-conditioned policy $\pi_g(\mathbf{a_t} \mid, \mathbf{s_t}, \mathbf{g_t})$ is uniquely determined. Therefore, we can define subtasks as sub-sequences that can: (1) be representative of common behavior patterns (because $\mathbf{g_t}$ guided the selection of $\mathbf{a_t}$) (2) avoid uncertainties in policy $\pi(\mathbf{a_t} \mid \mathbf{s_t})$, that is, different distributions of action predictions given a state. The formal definition goes as in Def. 5.

**Definition 5.** *Subtask* is defined as a set of $J$ options $\mathcal{O} = \{\mathcal{O}_j\}_{j=J}^J$, s.t. for some partition of trajectories, $\xi_p = \{\mathbf{s_t}, \mathbf{a_t}, ...\}_{t=1}^{\leq L}$ as a sub-sequence of $(\mathbf{s_t}, \mathbf{a_t})$ from any trajectory $\tau_n$ and $L$ is the maximum length of the sub-sequence, and also the maximum lag of each subtask:

$$\min \mid J \mid, \text{ s.t.}$$
$$(\forall \xi_p)\, (\exists \mathcal{O}_j)\, \xi_p \sim \mathcal{O}_j \text{ , and if } (\exists \mathbf{s_i} \in \xi_p, \xi_{p\prime}(p \neq p\prime)) \qquad (1)$$
$$\text{that } \pi_j(\mathbf{a_t} \mid \mathbf{s_t} = \mathbf{s_i}) \neq \pi_{j'}(\mathbf{a_t} \mid \mathbf{s_t} = \mathbf{s_i}), \text{ then } \mathbf{s_i} \sim \mathcal{O}_j, \mathcal{O}_{j'}, j \neq j'$$

We require every sub-sequence to be generated from some option $\sim \mathcal{O}_j$, which means that its first state $\xi_p(0) \in \mathcal{I}_j$ and the last state is a termination state $\beta_j(\xi_p(-1)) = 1$, and the actions in $\xi_p$ are generated by $\pi_j(\mathbf{a_t} \mid \mathbf{s_t})$. Importantly, when there are multiple policies available at one state $\mathbf{s_i}$, i.e. $\pi_j(\mathbf{a_t} \mid \mathbf{s_t} = \mathbf{s_i}) \neq \pi_{j'}(\mathbf{a_t} \mid \mathbf{s_t} = \mathbf{s_i})$, then these policies should correspond to different options, $\mathcal{O}_j$ and $\mathcal{O}_{j'}$ ($j \neq j'$), in order to avoid unmodeled uncertainty for the imitator.

By definition, subtasks are our way of recovering a minimal number of options from trajectories, and we view sub-sequences in the data as instantiations of options, because they are generated from corresponding SMDPs. Then, each $\tau_n$ should be *some* combination of those sub-sequences. On the one hand, we aim to avoid excessively granular partitions that result in a large number of one-step options, because it is essential to capture long-term patterns. On the other hand, we seek a sufficient number of subtasks to avoid policy ambiguity; when multiple options are available for a single state, we should be able to differentiate them by setting distinct subgoals. Consequently, in situations with varying distributions of action predictions, we employ different subtasks to capture these differences. This involves selecting subgoals to predict actions that a single policy cannot suffice. For example, determining to turn left at a crossroad as one subgoal and turning right as another.

**Justification of necessity** By this definition of subtasks, we reinforce the necessity of learning subtasks from the perspective of avoiding policy uncertainties in multiple trajectories. Since the trajectories are collected by a variety of human or robot experts under different tasks, they are likely to exhibit different optimal policies. If there is an ambiguity about which policy to imitate, i.e., there are multiple optimal action predictions at hand, we can comprehend it as there is a latent variable $\mathbf{g_t}$ affecting the prediction of $\mathbf{a_t}$, namely a subgoal that helps to select action. Learning one policy distribution $p(\mathbf{a_t} \mid \mathbf{s_t})$ from a mixture of different policies ignores the variety of behaviors, and focuses merely on the marginal policy $\pi(\mathbf{a_t} \mid \mathbf{s_t})$ rather than a subgoal conditioned policy $\pi_g(\mathbf{a_t} \mid \mathbf{s_t}, \mathbf{g_t})$, leading to unsatisfactory results. To our knowledge, we are the first ones to give a definition that eliminates the ambiguity in the definition of subtasks. While previous works provide only general and vague definitions by human intuition (McGovern and Barto [2001], Şimşek and Barto [2004], Wang et al. [2014], Paul et al. [2019], Krishnan et al. [2017], Kipf et al. [2019], Sharma

et al. [2019b], Yu et al. [2019]), e.g. semantically meaningful sub-trajectories, we explicitly express subtasks as patterns that capture *all policy uncertainties* exhibited in the dataset. These uncertainties are then mitigated by policy selections, i.e. the subgoal-conditioned policy.

### 3.3 Learning Subtasks with Seq-NMF

In Sec. 3.1, we give the conditions to identify subgoals as selections (verified in experiments in Sec. 5.1). In Sec. 3.2, we provide a formal definition of subtasks. Combining these concepts, we connect subgoals to subtasks: *subgoals* are multi-dimensional binary variables $\mathbf{G}_t \in \{0, 1\}^J$, while *subtasks* are sets of options $\mathcal{O}$ that generate sub-sequences $\xi_p = \{\mathbf{s_t}, \mathbf{a_t}, ...\}_{t=1}^{\leq L}$. In particular, we use $\mathbf{G}_t^{(j)}(\xi_p) = \mathbb{1}(\xi_p \sim \mathcal{O}_j)$ to indicate whether the subgoal of a sub-sequence is generated from an SMDP captured by $\mathcal{O}_j$, where $\mathbb{1}(\cdot)$ is the identify function. Similarly, we use $\mathbf{H}_t^{(j)}([\mathbf{s_t}, \mathbf{a_t}]) = \mathbb{1}(\mathbf{s_t} \in \mathcal{I}_j)$ to indicate whether $\mathbf{s_t}$ is an initial state of $\mathcal{O}_j$.

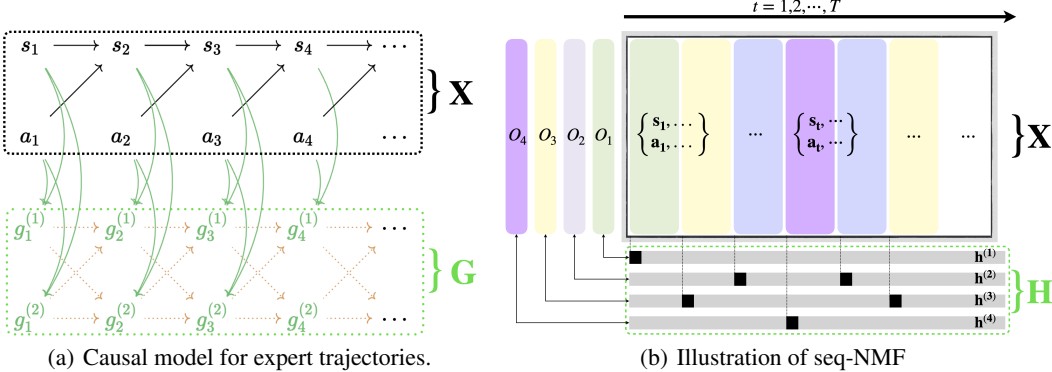

(a) Causal model for expert trajectories.    (b) Illustration of seq-NMF

Figure 3: Figure (a) is the causal model for expert trajectories, which is further abstracted as the matrices in Figure (b), which can be learned by a seq-NMF algorithm. In both figures, data matrix $X$ is the aggregated $\{\mathbf{s_t}; \mathbf{a_t}\}_{t=1}^T$, and $\mathbf{H} \in \{0, 1\}^{J \times T}$ represents the binary subgoal matrix.

This formulation can be intuitively understood as follows: subtasks represent the behavior patterns one might select to perform and are thus exhibited in the expert trajectories. In contrast, the subgoal is the selection variable itself, indicating whether or not this behavior pattern has been executed.

**Method: Sequential Non-Negative Matrix Factorization**    Learning such a binary coefficient matrix and feature pattern is closely related to the non-negative matrix factorization (NMF) (Lee and Seung [1999]), which focuses on decomposing a complex dataset into a set of simpler, interpretable components while maintaining non-negativity constraints. A thorough review of NMF is in Appx. G.

In our setting, instead of using a vector to represent a pattern, we want the pattern to be temporally extended, sharing the merit of those works of extensions of NMF (Smaragdis [2004, 2006], Mackevicius et al. [2019]). We set $\mathbf{x_t} = (\mathbf{s_t}; \mathbf{a_t})$ and concatenate $\mathbf{x_t}$ across time to form a data matrix $X$. We then transform the problem of learning subgoals into a matrix factorization problem: identifying the repeated patterns within sub-sequences. The entire data matrix (trajectories) can be reconstructed with *subtasks* representing temporally extended patterns, and binary indicator representing which option is selected at each time step. We define $\mathbf{O} = [\ \mathbf{O}_1\ \ \mathbf{O}_2\ \ \cdots\ \ \mathbf{O}_J\ ] \in \mathbb{R}^{D \times J \times L}$ as a three-dimensional tensor, with $J$ subtask patterns $\mathbf{O}_j \in \mathbb{R}^{D \times L}$, and $\mathbf{H} = [\ \mathbf{h}_1\ \ \mathbf{h}_2\ \ \cdots\ \ \mathbf{h}_T\ ] \in \{0, 1\}^{J \times T}$ as corresponding indicator binary matrix, where $D = d_s + d_a$ is the dimension of $\mathbf{x_t}$, and $L$ is the maximum length of each subtask pattern. We construct the decomposition as:

$$\mathbf{X} \approx \mathbf{O} * \mathbf{H}, \text{where}(\mathbf{O} * \mathbf{H})_{dt} = \sum_{j=1}^{J} \sum_{\ell=0}^{L-1} \mathbf{O}_{dj\ell} \mathbf{H}_{j(t-\ell)}, \tag{2}$$

and $*$ is a convolution operator that aggregates the patterns across time lag $L$. Then the optimization problem is transformed into:

$$(\mathbf{O}^*, \mathbf{H}^*) = \underset{\mathbf{O}, \mathbf{H}}{\arg\min} \left( \|\widetilde{\mathbf{X}} - \mathbf{X}\|_F^2 + \mathcal{R} \right)$$

$$s.t. \quad \widetilde{\mathbf{X}}_{dt} = \sum_{j=1}^{J} \sum_{\ell=0}^{L-1} \mathbf{O}_{dj\ell} \mathbf{H}_{j(t-\ell)}. \tag{3}$$

$\|\cdot\|_F$ is the Frobenius norm, and $\mathcal{R}$ is the regularizor. In order for it to fit our framework proposed in Sec. 3.1 and Sec. 3.2, we need three terms in the regularizor: $\mathcal{R} = \mathcal{R}_{\mathrm{bin}} + \mathcal{R}_1 + \mathcal{R}_{\mathrm{sim}}$ with corresponding learning rates $\lambda_{\mathrm{bin}}$, $\lambda_1$ and $\lambda_{\mathrm{sim}}$.

The first term $\mathcal{R}_{\mathrm{bin}}$ corresponds to *the binary nature of selection*. Because the set $\{0,1\}$ is not convex, we remove the constraint $\mathbf{H}_t^{(j)} \in \{0,1\}$ to a regularizer $\mathcal{R}_{\mathrm{bin}} = \lambda_{\mathrm{bin}}\|\mathbf{H} \odot (1-\mathbf{H})\|_2^2$, forcing the subgoal to be binary, where $\odot$ is the element-wise product. Because of *sparsity of subtasks* in its definition, we require a minimal number of subgoals, which should be an L0 penalty term. We use the L1 penalty $\mathcal{R}_1 = \lambda_1\|\mathbf{H}\|_1$ to approximate such sparsity since solving the L0 regularized problem is NP-hard. Finally, we should have *distinct features of subtasks*: distinct common patterns should be distinguished as the same subtask, i.e. there should not be similar or duplicated patterns between any two different subtasks $\mathbf{O}_j$ and $\mathbf{O}_{j'}$. We use $\mathcal{R}_{\mathrm{sim}} = \lambda_{\mathrm{sim}} \left\| \left(\mathbf{O} \overset{\leftarrow}{*} \mathbf{X}\right) \mathbf{S} \mathbf{H}^\top \right\|_{1, i \neq j}$ to avoid such redundancy, where $\mathbf{S}$ is a $T \times T$ smoothing matrix: $\mathbf{S}_{ij} = 1$ when $|i-j| < L$ and $\mathbf{S}_{ij} = 1$ otherwise. Specifically, the first term $\left(\mathbf{O} \overset{\leftarrow}{*} \mathbf{X}\right)$ calculates the overlap of data $X$ with subtask pattern $j$ at each time step $t$, where $\left(\mathbf{O} \overset{\leftarrow}{*} \mathbf{X}\right)_{jt} = \sum_{\ell=0}^{L-1} \sum_{d=1}^{D} \mathbf{O}_{dj\ell}\mathbf{X}_{j(t+\ell)}$ and $\overset{\leftarrow}{*}$ is the transposed convolution operator. Then by multiplying the loadings $\mathbf{S}\mathbf{H}^\top$ within the time shift of $L$, we obtain the correlation between different patterns' overlapping with the data. If the correlation is high, it means that the two patterns have similar power in explaining the data at time $t$. Diagonal entries are omitted. We provide the detailed discussion on subtasks ambiguity in Appx. F.

**Optimization.** In terms of optimization, we derive multiplication rules which have been proven to be more efficient in solving problems with non-negative constraints ( Lee and Seung [1999]), rather than relying on standard gradient descent. The detailed derivation is provided in Appx. G and the overall algorithm of seq-NMF is described in Appx. C.1.

$$\mathbf{O}_{..\ell} \leftarrow \mathbf{O}_{..\ell} \cdot \frac{\mathbf{X} * \mathbf{H}^\top}{\widetilde{\mathbf{X}} * \mathbf{H}^\top + \frac{\partial \mathcal{R}}{\partial \mathbf{O}_\ell}}, \quad \mathbf{H} \leftarrow \mathbf{H} \cdot \frac{\mathbf{O} \overset{\leftarrow}{*} \mathbf{X}}{\mathbf{O} * \widetilde{\mathbf{X}} + \frac{\partial \mathcal{R}}{\partial \mathbf{H}}} \tag{4}$$

# 4 Transfering to New Tasks

After learning subtasks from demonstrations, it is intuitive to utilize them by augmenting the action space with the subgoal selection. We learn a new policy that takes in both the state and subgoal as input, same as in other literature (Sharma et al. [2019b], Kipf et al. [2019], Jing et al. [2021], Jiang et al., Chen et al. [2023]).

Among all the different ways to perform IL, such as Behavioral Cloning (BC) (Bain and Sammut [1995]), Inverse Reinforcement Learning (IRL) (Abbeel and Ng [2004], Ng et al. [2000], Ziebart et al. [2008], Finn et al. [2016]), and Generative Adversarial Imitation Learning (GAIL) (Ho and Ermon [2016], Fu et al. [2018]), we adopt a GAIL-based approach that matches the occupancy measure between the learned policy and the demonstrations through adversarial learning to seek the optimal policy. The overall objective is:

$$\min_{\pi_g} \max_{\theta} \mathbb{E}_{(s_t, a_t, g_t) \sim \tau} \log(1 - D_\theta(s_t, a_t, g_t)) + \mathbb{E}_{(\tilde{s}_t, \tilde{a}_t, \tilde{g}_t) \sim \pi_g} \log(D_\theta(\tilde{s}_t, \tilde{a}_t, \tilde{g}_t)), \tag{5}$$

where $\pi_g$ is the augmented policy, and $D_\theta : \mathcal{S} \times \mathcal{A} \times J \to [0,1]$ is a parametric discriminator that aims at distinguishing between the samples generated by the learned policy and the demonstrations. The policy is trained via PPO (Schulman et al. [2017]). Algorithms for the overall training procedure and the execution of policy can be found in Appx. C.2. Comparison with other IL algorithms is provided in our experiment in Sec. 5.3.

# 5 Experiments

The goals of our experiments are closely tied to our formulation in Sec. 3. In Sec. 5.1, we verify the existence of selection in data. In Sec. 5.2, we evaluate the effectiveness of seq-NMF in recovering subgoals as selections. In Sec. 5.3, we demonstrate the learned subgoals and subtasks are transferable to new tasks by re-composition.

## 5.1 Verifying Subgoals as Selections

We first verify the theory in Sec. 3 that selections can be identified in data by the following CI tests: (1) whether $s_t \not\perp\!\!\!\perp a_t \mid g_t$ holds, (2) whether $g_t \not\perp\!\!\!\perp a_{t+1} \mid g_t$ holds, and (3) whether $s_{t+1} \perp\!\!\!\perp g_t \mid s_t, a_t$ holds. Our empirical results provide an affirmative answer to all these questions, suggesting that selections do exist, and they can serve as subgoals.

**Synthetic Color Dataset** We follow the didactic experiment in Jiang et al. and construct color sequences similarly. The dataset consists of repeating patterns of repetitive color, either with 3 or 10 steps of time lag in each pattern, of which Fig. 4 is an illustration. Details about the construction and the CI test results are elaborated in Appx. D.1.

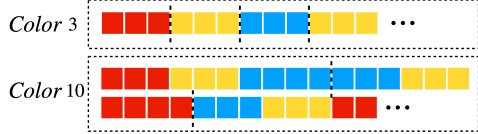

Figure 4: Patterns in $Color$-3 and -10.

**Driving Dataset** In the driving environment, there are two tasks to finish. As shown in Fig. 5, both cars start at the left end, either facing up or down. The first task is to drive to the right end following the yellow path, while the second task is to follow the blue path. Each state is represented by a tuple $(x, y, \theta) \in \mathbb{R}^3$ (coordinates and orientation), and each action is the angular velocity $\Delta\theta \in \mathbb{R}$. We collected 100 trajectories in total, 50 for each task.

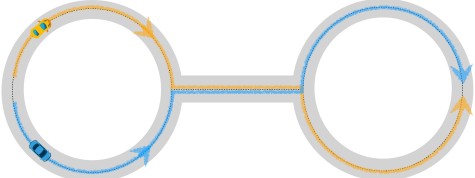

Figure 5: Two tasks in Driving environment.

| CI test | Single-step | Multi-step |
|---|---|---|
| (1) $s_t \perp\!\!\!\perp a_t \mid g_t$ | 0.00 | 0.00 |
| (2) $g_t \perp\!\!\!\perp a_{t+1} \mid g_{t+1}$ | $1e^{-7}$ | 0.008 |
| (3) $s_{t+1} \perp\!\!\!\perp g_t \mid s_t, a_t$ | 0.40 | 0.65 |

Table 1: P-values for CI tests in Driving.

**CI test results** We show our results in Tab. 1 for the Driving dataset. In short, the answer for (1) whether $s_t \not\perp\!\!\!\perp a_t \mid g_t$, (2) whether $g_t \not\perp\!\!\!\perp a_{t+1} \mid g_t$, and (3) $s_{t+1} \perp\!\!\!\perp g_t \mid s_t, a_t$ are all yes. For Driving Dataset, as is shown in Tab. 1, single-step denotes that we are treating $s_t$ at different time steps as different variables, and calculate the mean of p-values across time. Multi-step denotes that we are aggregating $s_t$ at different times steps together as a single variable, and sample a subset of data every time, calculate the mean of the p-values across subsets. Note that all the variance of p-values in Tab. 1 are less than 0.001, so we only record the mean here. By testing CI condition (1) as sufficient, and CI conditions (1) and (2) together as both necessary and sufficient, we verify that subgoals are essentially selection variables. Moreover, by CI condition (3), we also validate that the next state should be determined solely by the previous state and action, and particularly depends on the physical transition, but not influenced by the subgoals.

## 5.2 Evaluating the Effectiveness of Seq-NMF

Next, we evaluate the seq-NMF in recovering selections and discovering subtasks. Results in the Driving dataset are in Fig. 6 while those for $Color$ are in Appx. D.4. The y-axis represents the dominance of each subtask in explaining the whole sequence. We plot two sequences and each lasts for around 110 steps. Our algorithm finds the "crossing point" and automatically partitions every trajectory into three subtasks (before reaching the first crossing point, in the middle, and after reaching the second crossing point).

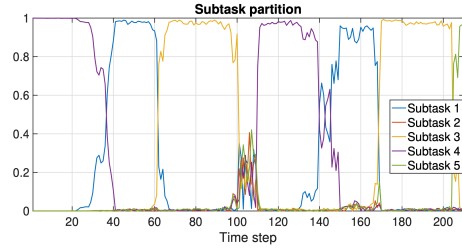

Figure 6: seq-NMF result on Driving.

## 5.3 Transfering to New Tasks

**Kitchen Dataset**   The evaluation of the imitation learning is
performed on a challenging Kitchen environment from D4RL(Fu
et al. [2020]) benchmark. Each agent is required to perform a
sequence of subtasks including manipulating the microwave,
kettle, cabinet, etc. Each task is composed of 4 subtasks and the
composition is unknown to the agent. We use the demonstrations
provided by (Gupta et al. [2019]) for reproducibility, which
only contain state and action pairs but not reward. We use the
demonstrations of two tasks for training, and require the agent
to accomplish a target task that has a different composition of
subtasks than any one of the demonstrations, but the units of
subtasks have been performed. Detailed description is included in Appx. D.2.

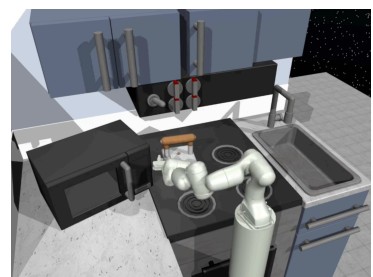

Figure 7: Kitchen environment

**Baseline methods**   We compare our method with the following state-of-the-art (SOTA) hierarchical
imitation learning methods to prove its efficacy: (1) **H-AIRL** (Chen et al. [2023]) is a variant of the
meta-hierarchical imitation learning method proposed in Chen et al. [2023] that doesn't incorporate
the task context, and is learning an option-based hierarchical policy just as in our setting. (2)
**Directed-info GAIL**(Sharma et al. [2019b]) and (3) **Option-GAIL** (Jing et al. [2021]) are two other
competitive baselines that are proven to be effective in solving multi-task imitation learning.

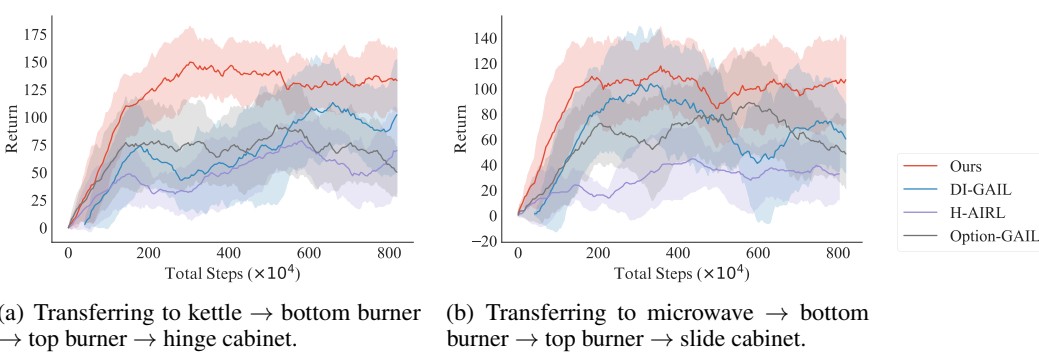

(a) Transferring to kettle → bottom burner
→ top burner → hinge cabinet.

(b) Transferring to microwave → bottom
burner → top burner → slide cabinet.

Figure 8: Results for solving new tasks in the Kitchen environment w.r.t. training steps.

**Results on new tasks with 4-subtasks**   We show the results on new tasks with 4-subtasks in Fig. 8.
We use the episodic accumulated return as the metric. Training is repeated 5 times randomly for
each algorithm, with the mean shown as a solid line and the standard deviation as a shaded area.
We observe that our method outperforms all the baselines in both tasks. The agent trained with
the selection-based subtasks can quickly adapt to the new task with different subtask compositions,
achieving a higher return. The results show that our method can effectively transfer the subtask
knowledge learned from the demonstrations to new tasks.

**Results on new tasks with 5-subtasks
(generalized)**   We conduct additional
experiments by considering a distribu-
tion shift problem that involves longer-
horizon tasks, and plot the results in
Fig. 9. Specifically, under the Kitchen
environment, we keep the same train-
ing set of tasks (each task is composed
of 4 sequential manipulation subtasks),
and tests the method's generalizability
to a new task with different permutation
of one more subtask, i.e. 5 sequential
manipulation subtasks. Such generaliza-
tion to longer-horizon tasks is not taken into consideration by some of the existing works (Chen

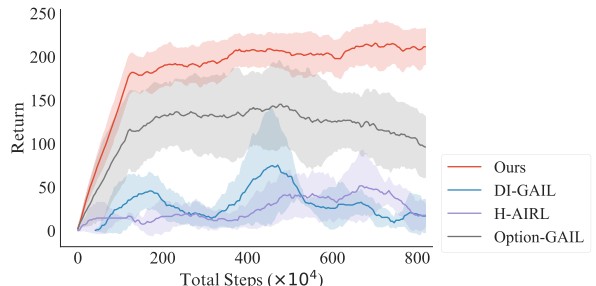

Figure 9: (Generalized) Transfering to microwave → bot-
tom burner → top burner → slide cabinet → hinge cabinet.

et al. [2023]), and our empirical results show that our formulations are able to deal with such a more challenging distribution shift problem.

By detecting the existence of selections in data, and recognizing them as subgoals, we can effectively learn the subtasks that are useful for future use–subgoals are the selection indicators for the subtask patterns. Such a procedure is in contrast to other methods that neglect the real structure in data and merely adopt a maximization of the likelihood objective, which might be misleading for recovering meaningful patterns. Also, their joint optimization of up to five networks (Chen et al. [2023]) makes model instability a major concern. Our method, on the other hand, learns subgoals directly from demonstrations, disentangling it from the policy training. This makes the learning more stable and interpretable, and the learned subtasks can easily adapt to new scenarios.

## 6  Conclusion

In short, we target at a subtask decomposing problem. While previous research has not sufficiently analyze the concept of subtasks, which might lead to an inappropriate inference of subgoals, we propose to view subtasks as outcomes of selections. We first verify the existence of selection variables in the data based on our theory. From this perspective, we recognize subgoals as selections and develop a sequential non-negative matrix factorization (seq-NMF) method for subtask discovery. We rigorously evaluate the algorithm on various tasks and demonstrate the existence of selection and the effectiveness of the method. Finally, our empirical results in a challenging multi-task imitation learning setting further show that the learned subtasks significantly enhance generalization to new tasks, suggesting exciting directions on uncovering the causal process in the data, also showing a new perspective on improving the transferability of policy.

The main limitation in this work lies in that we are not yet able to deal with the scenarios where there might be multiple factors at work, such as the case where there are both underlying confounders and selections. Confounders might be related to other types of distribution shift, e.g. change in the system dynamics, robot embodiment, etc. In future work, we will investigate the causal process in other contexts, and aiming at providing a more general framework for subtask discovery.

## Acknowledgments and Disclosure of Funding

We thank Clark Glymour, Xiangchen Song, Biwei Huang, Zeyu Tang, Guangyi Chen, Jiaqi Sun, Yiding Jiang, for the valuable discussions and their support, and we thank the anonymous reviewers for their suggestions. This material is based upon work supported by NSF Award No. 2229881, AI Institute for Societal Decision Making (AI-SDM), the National Institutes of Health (NIH) under Contract R01HL159805, and grants from Salesforce, Apple Inc., Quris AI, and Florin Court Capital. Funding to attend this conference was provided by the CMU GSA/Provost Conference Funding.

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

# A  Related Works

**Skill or option discovery**   While there are multiple terms across several lines of research, e.g. skills or options, they all refer to the same problem and focus on learning a set of subtasks that can be used to solve a complex task via hierarchical planning. The problem is also our main focus here. The subgoals are defined heuristic as bottle-neck regions in tasks, for example, by finding states that are visited frequently by the expert ( McGovern and Barto [2001], Şimşek and Barto [2004]), finding a minimal cut of the state transition graph ( Şimşek et al. [2005]), or by clustering and other similarity measures ( Wang et al. [2014], Paul et al. [2019]). Recently, the skills are learned through modeling skills as latent variables and maximizing likelihood. For example, there are deep option learning methods ( Krishnan et al. [2017], Fox et al. [2017], Bera et al. [2020]), hierarchical clustering ( Zhu et al. [2022]), information theory-induced methods ( Sharma et al. [2019a], Lee [2020]), segmentation and abstractions modeling ( Kipf et al. [2019], Tanneberg et al. [2021]), and methods that leverage other criteria such as minimal description length ( Zhang et al. [2021], Jiang et al.). However, these methods do not provide a clear definition of skills or options, and the learned partitions face challenges in their interpretability. We can address the problem by identifying sub-goals as selections that enhance interpretability.

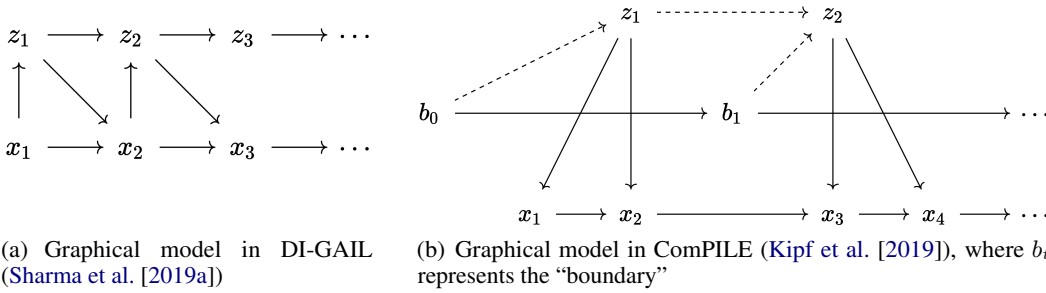

(a) Graphical model in DI-GAIL (Sharma et al. [2019a])

(b) Graphical model in ComPILE (Kipf et al. [2019]), where $b_t$ represents the "boundary"

Figure 10: Two examples of the graphical models used in other literature.

For example, Fig. 10 shows two different graphical models proposed in DI-GAIL and ComPILE that are representative. State and action are aggregated into one single variable $x_t \coloneqq \{s_t, a_t\}$, and the subtask variable is denoted as $z_t$ in both cases. Their corresponding methods do not take into consideration of the true generation process, thus might be unable to reflect the real data structure and lead to biased inference.

**Multi-task IL (MIL)**   Current MIL methods aim at training a policy that can be conveniently adapted to multiple tasks, often done so by incorporating a task context variable to condition on, such as (Seyed Ghasemipour et al. [2019], Yu et al. [2019]). However, the task variable is not always available and these methods neglect the hierarchical structure in accomplishing the task. Then Chen et al. [2023] takes into account skill/option framework to build a hierarchical policy along with the context variable. However, this approach inherently suffers from optimization problems and require demanding hyperparameter tuning process, due to the overall complexity of the framework, while we provide a solution with disentangled optimization procedures that is efficient to deploy.

We also build a rich literature in the following topics, for which our insights provide important implications:

**Causal RL and IL**   This line of research introduces causal ideas to improve IL problems, by conditioning the imitation policy on the causal parents (De Haan et al. [2019]) or by learning a compact causal structure between the states, actions and rewards (Lee et al. [2021]). However, the selection structure under this context is still not sufficiently investigated, and we provide this new perspective.

**Selection bias**   In causal literature, previous works mainly focus on understanding selection as a distortion of data, and aim at alleviate the selection effect (Spirtes et al. [1995], Hernán et al. [2004], Zhang [2008], Bareinboim et al. [2014], Zhang et al. [2016], Correa et al. [2019], Forré and Mooij [2020], Versteeg et al. [2022], Chen et al. [2024]). However, though controlling the selection bias is important, they fail to view the selection as a source of information that can facilitate learning and inferencing. Zheng et al. [2024] propose methods to discover the selection structure in the sequential data. We explore the underlying selection structure in imitation learning settings and fill the missing gaps in comprehensively understanding the selection process in the data.

# B Proofs

## B.1 Proof of Proposition 1

**Proposition 1.** (Sufficient condition) Assuming that the graphical representation is Markov and faithful to the measured data, if $\mathbf{s_t} \not\perp\!\!\!\perp \mathbf{a_t} \mid \mathbf{d_t}$, then $\mathbf{d_t}$ is a selection variable, i.e., $\mathbf{d_t} := \mathbf{g_t}$, under the assumption that:

1. (confounder, selection, and intermediate nodes can not co-exist) At each time step, $\mathbf{d_t}$ can only be one of $\{\mathbf{c_t}, \mathbf{g_t}, \mathbf{m_t}\}$. (For a relaxation of this assumption, see Appx. B.4).

2. (consistency in a time series) At every time step, $\mathbf{d_t}$ plays the same role as one of $\{\mathbf{c_t}, \mathbf{g_t}, \mathbf{m_t}\}$.

We can use d-separation Pearl [2009] to distinguish the $\mathbf{d}_t := \mathbf{g}_t$ case from the other two kinds of dependencies. $\mathbf{s}_t \not\perp\!\!\!\perp_d \mathbf{a_t} \mid \mathbf{d}_t$ if $\mathbf{d}_t := \mathbf{g}_t$ because there is a path $\mathbf{s_t} \to \mathbf{g_t} \leftarrow \mathbf{a_t}$, so $\mathbf{s}_t \not\perp\!\!\!\perp \mathbf{a_t} \mid \mathbf{d}_t$. But if $\mathbf{d_t} := \mathbf{c}_t$ or $\mathbf{d}_t := \mathbf{m}_t$, because $\mathbf{s_t} \perp\!\!\!\perp_d \mathbf{a_t} \mid \mathbf{c_t}$ in the confounder case and $\mathbf{s_t} \perp\!\!\!\perp_d \mathbf{a_t} \mid \mathbf{m_t}$ in the intermediate node case, then $\mathbf{s}_t \perp\!\!\!\perp \mathbf{a_t} \mid \mathbf{d}_t$.

## B.2 Proof of Proposition 2

**Proposition 2.** (Necessary and sufficient condition) $\mathbf{d_t}$ is a selection variable ($\mathbf{d_t} := \mathbf{g_t}$) if and only if $\mathbf{s_t} \not\perp\!\!\!\perp \mathbf{a_t} \mid \mathbf{d_t}$ and $\mathbf{d_t} \not\perp\!\!\!\perp \mathbf{a_{t+1}} \mid \mathbf{d_{t+1}}$.

In Prop. 1, we have already proved that $\mathbf{s_t} \not\perp\!\!\!\perp \mathbf{a_t} \mid \mathbf{d_t}$ is sufficient to show that $\mathbf{d_t} := \mathbf{g_t}$. For the necessary part, we need to show that if $\mathbf{d}_t$ is selection, then it entails that $\mathbf{d_t} \not\perp\!\!\!\perp \mathbf{a_{t+1}} \mid \mathbf{d_{t+1}}$. Because $\mathbf{g_t} \not\perp\!\!\!\perp_d \mathbf{a_{t+1}} \mid \mathbf{g_{t+1}}$, so by d-separation, $\mathbf{g_t} \not\perp\!\!\!\perp \mathbf{a_{t+1}} \mid \mathbf{g_{t+1}}$.

## B.3 Proof of Proposition 3

**Proposition 3.** (Necessary condition) If $\mathbf{d_t}$ is a selection variable ($\mathbf{d_t} := \mathbf{g_t}$), then $\mathbf{s}_{t+1} \perp\!\!\!\perp \mathbf{g}_t \mid \mathbf{s}_t, \mathbf{a}_t$. (Such independency does not hold true for confounders case which is discussed in Appx. B.3)

When conditioning on $\mathbf{s_t}$ and $\mathbf{a_t}$, then $\mathbf{g_t}$ and $\mathbf{s_{t+1}}$ are d-separated. But for a confounder case, $\mathbf{c_t}$ and $\mathbf{s_{t+1}}$ are not. The difference is essentially because that we have $\mathbf{g_t} \to \mathbf{g_{t+1}} \leftarrow \mathbf{s_{t+1}}$ where there is an unshielded collider $\mathbf{g_{t+1}}$ between $\mathbf{g_t}$ and $\mathbf{s_{t+1}}$ that blocks the path, while $\mathbf{c_t} \to \mathbf{c_{t+1}} \to \mathbf{s_{t+1}}$ is not blocked. Therefore, $\mathbf{s_{t+1}} \perp\!\!\!\perp \mathbf{g_t} \mid \mathbf{s_t}, \mathbf{a_t}$, but $\mathbf{s_{t+1}} \not\perp\!\!\!\perp \mathbf{c_t} \mid \mathbf{s_t}, \mathbf{a_t}$.

## B.4 Relaxation of Assumptions

We extend our theory in Sec. 3.1 by two types of relaxations of our assumptions. We provide a more general graphical model that represents the data generation process in Fig. 11. There are two parts that will be discussed in this section: (1) the inclusion of co-existence of $\mathbf{g_t}$, $\mathbf{c_t}$ and $\mathbf{m_t}$ and (2) the higher-order structure behind those variables (e.g. the set of higher-order confounders and selections are denoted as $U_C$ and $U_S$, respectively, in Fig. 11).

### B.4.1 Co-existance of $\mathbf{c_t}$, $\mathbf{g_t}$ and $\mathbf{m_t}$

The first point we assert in this section is that it is possible $\mathbf{d}_t$ does not take only *one* of $\{\mathbf{c_t}, \mathbf{g_t}, \mathbf{m_t}\}$. In the main paper, we only discuss the pure case for simplicity, however, it is also possible that $\mathbf{d}_t$ can take *multiple* characters of $\{\mathbf{c_t}, \mathbf{g_t}, \mathbf{m_t}\}$. We argue that even if the data is generated under multiple hidden variables, for example, both confounders and selections are at play, our sufficiency criteria for recognizing the existence of selection still holds, and propose the following Prop. 4.

**Proposition 4.** (Sufficient condition under multiple types of hidden variables) Assuming that the graphical representation is Markov and faithful to the measured data, if $\mathbf{s_t} \not\perp\!\!\!\perp \mathbf{a_t} \mid \mathbf{d_t}$, then $\mathbf{g_t}$ must exist in the hidden structure, i.e., $\mathbf{g_t} \in \mathbf{d_t}$ (we use $\mathbf{d_t}$ to denote the set of hidden variables), under the modified assumptions that :

1. (one or more structures of confounder, selection, and intermediate node) At each time step, $\mathbf{d_t}$ can be a subset of $\{\mathbf{c_t}, \mathbf{g_t}, \mathbf{m_t}\}$. i.e. $\mathbf{d_t} \subseteq \{\mathbf{c_t}, \mathbf{g_t}, \mathbf{m_t}\}$.

2. (consistency in a time series) Similar as in Prop. 1.

**Proof** In the graphical model of Fig. 11, both confounders ($\{\mathbf{c_t}\}$) and selections ($\{\mathbf{g_t}\}$) are exhibited. Then conditioning on $\mathbf{d_t}$ does not change the key CI criteria we proposed in Prop. 1: $\mathbf{s_t} \not\perp\!\!\!\perp \mathbf{a_t} \mid \mathbf{d_t}$, because as long as the path $\mathbf{s_t} \to \mathbf{g_t} \leftarrow \mathbf{a_t}$ is d-connected (when two variables are not d-separated, they are d-connected) when conditioning on $\mathbf{g_t}$, the dependency between $\mathbf{s_t}$ and $\mathbf{a_t}$ is preserved, regardless of whether there are confounders or intermediate nodes. If $\mathbf{g}_t$ is not present, i.e. $\mathbf{d}_t \subseteq \{\mathbf{c}_t, \mathbf{m}_t\}$, then there is no dependency between $\mathbf{s_t}$ and $\mathbf{a_t}$ when conditioning on $\mathbf{d_t}$.

**Implications** Such a relaxation that includes multiple types of hidden variables is reasonable in real-world scenarios. While the observed data is generated by a subgoal-conditioned policy, it is still possible that there is some hidden confounder that influences the joint data distribution. For example, when you are selecting which way to go to school, you may be influenced by the weather. Given the same state and action, say, you decide to walk out of home, changing the weather will change the distribution of the next state: it causes you to act otherwise, e.g. stay at home. On the other hand, changing your actions would not influence the weather. In this case, the weather is a confounder that is distinct from the subgoal, but still influences the data generation process. The main proposition Prop. 1 is one that provides the sufficient condition to identify selections, and Prop. 4 implies that we could not eliminate the possibility of confounders and intermediates at play together with selections. Learning the hidden confounder or intermediate structures along with selections is another exciting topic that we leave for future work.

### B.4.2  Higher-order Structure Behind Data

In the main paper, we assume that the true causal graph of the data takes one of the three scenarios in Fig. 2, with no hidden high-level structures. A second relaxation we make is the inclusion of higher order structures. We argue that if there are higher-order structures in the data, and loosen the assumption of a direct adjacency between $\mathbf{d_t}$ and $\mathbf{d_{t+1}}$, then Prop. 1 still holds true.

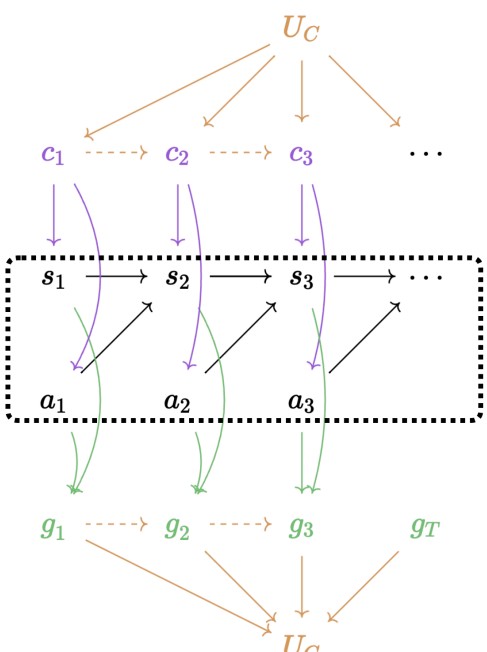

**Illustration** Take the case of $U_G$ (the set of higher-order goals) for example. The idea that there potentially could be hierarchical subgoals is natural: we take the example of `commuting to New York` in Sec. 1 as an illustration. The first subtask `walking out of the house` is further decomposed as `standing up`, `grabbing the luggage` and `walking to the door`. Each one of the four behaviors can also be understood as smaller subtasks that involve multiple more primitive actions: for walking, you need to move your left leg, right hand, then right leg, left hand, etc. On a grander scale, `commuting to New York` could possibly be a part of a bigger task of *going on a business trip*: `commuting to New York`, `going to a meeting`, `giving a presentation`. In this example, `walking to the door` would be the first level of subgoals ($\mathbf{g_t}$). Then it contributes to a higher level of subgoal: `commuting to New York`, which we represent as $\dot{\mathbf{g}}_\mathbf{t} \in U_G$. Then this second-order subgoal further contributes to a third-order subgoal, the final goal of *going on a business trip*, which we represent as $\ddot{\mathbf{g}}_\mathbf{t} \in U_G$. All these subgoals has a pattern of $\mathbf{g_t} \to \dot{\mathbf{g}}_\mathbf{t} \to \ddot{\mathbf{g}}_\mathbf{t} \to \cdots$, where the higher levels of subgoals are always selective of lower levels of subgoals (achieving the lower-order goals causes us to achieve the higher-order goals). The consecutive subgoals are dependent of each other because of conditioning on higher-order subgoals: $\mathbf{g_t} \not\perp\!\!\!\perp \mathbf{g_{t+1}} \mid \dot{\mathbf{g}}_\mathbf{t}$, because conditioning on $\dot{\mathbf{g}}_\mathbf{t}$ will make $\mathbf{g}_t$ and $\mathbf{g_{t+1}}$ d-connected. Noting that this relaxation of high-order structures is practical in the real-world setting, we provide a corresponding proposition in Prop. 5:

Figure 11: Graphical model with relaxed assumptions. (1) We allow the co-existance of $\mathbf{g_t}$ and $\mathbf{c_t}$ (similarly for $\mathbf{g_t}$ and $\mathbf{m_t}$ ). (2) We also assume there are potential higher-order underlying confounders $U_C$ in the data that create the dependencies between $\mathbf{c_t}$ and $\mathbf{c_{t+1}}$, and underlying selections $U_S$ in the data that create the dependencies between $\mathbf{g_t}$ and $\mathbf{c_{t+1}}$.

**Proposition 5.** (Higher-order structure) Assuming that there are higher-order structures behind $\mathbf{c_t}$, $\mathbf{g_t}$ and $\mathbf{m_t}$, and direct edges between $\mathbf{d_t}$ and $\mathbf{d_{t+1}}$ are not necessarily at present. Then Prop. 1 is still valid.

**Proof** The higher-order structure does not change the key CI criteria we proposed in Prop. 1 and Prop. 2. When conditioning on $\mathbf{d_t}$, if $\mathbf{d_t}$ is a confounder or intermediate, then it d-separates $\mathbf{s_t}$ and $\mathbf{a_t}$ and we will obtain $\mathbf{s_t} \perp\!\!\!\perp \mathbf{a_t}$. Only when $\mathbf{d_t}$ is a selection/subgoal, it would be a collider on the path between $\mathbf{s_t}$ and $\mathbf{a_t}$. Thus, only in the case of a selection, we will obtain $\mathbf{s_t} \not\perp\!\!\!\perp \mathbf{a_t}$. Now we have proved that the sufficient condition in Prop.(1) still holds.

This relaxation for the high-order structure implies that the direct adjacency between $\mathbf{d_t}$ and $\mathbf{d_{t+1}}$ is not necessary for the sufficiency criteria to hold.

# C Algorithms

## C.1 Algorithm for Seq-NMF

---
**Algorithm 1** SeqNMF for learning subtasks

---
**Input:** Data matrix $\mathbf{X}$, number of subtasks $J$, maximum time delay in a subtask $L$, regularization strength $\lambda_{\text{bin}}, \lambda_1, \lambda_{\text{sim}}$
**Output:** $\mathbf{O}$, $\mathbf{H}$
  Normalized $\mathbf{X}$ into $[0, 1]$
  Initialize $\mathbf{O}$ and $\mathbf{H}$ randomly
  Set $i = 1$
  **while** ($i \leq$ maxIter) and ($\Delta$ cost $\geq$ tolerance) **do**
    Update $\mathbf{H}$ using multiplicative update from Eqn. (4)
    Renormalize $\mathbf{H}$ so maximum value of $\mathbf{H}$ is 1, and normalize $\mathbf{O}$ accordingly.
    Update $\mathbf{O}$ using multiplicative update from Eqn. (4)
    $i = i + 1$
  **end while**
  Set $\lambda_{\text{bin}}, \lambda_1, \lambda_{\text{sim}}$ to zero, do one final unregularized multiplicative update of $\mathbf{O}$ and $\mathbf{H}$
**Return:** $\mathbf{O}$, $\mathbf{H}$

---

## C.2 Algorithms for Transfering to New Tasks

---
**Algorithm 2** Imitation Learning in new tasks with subgoals

---
**Input:** Subtask patterns $\mathbf{O}$, expert demonstrations $\{\mathbf{s_t}, \mathbf{a_t}, \cdots\}$.
  Initialize the policy $\pi_g$, discriminator $D_\theta$.
  **for** each training episode **do**
    Generate $M$ trajectories $\{\tilde{\mathbf{s}}_t, \tilde{\mathbf{a}}_t, \tilde{\mathbf{g}}_t\}$ by exploring in the target environment with $\pi_g$ and with Algo. 3.
    Update $D_\theta$ by minimizing $\mathcal{L}_{IL}$ based on $\{\mathbf{s}_t, \mathbf{a}_t, \mathbf{g}_t\}$ and $\{\tilde{\mathbf{s}}_t, \tilde{\mathbf{a}}_t, \tilde{\mathbf{g}}_t\}$
    Train $\pi_g$ by PPO (Schulman et al. [2017]), based on $\{\tilde{\mathbf{s}}_t, \tilde{\mathbf{a}}_t, \tilde{\mathbf{g}}_t\}$ and $D_\theta$ which defines the reward $R_{IL}$.
  **end for**

---

---
**Algorithm 3** Executing subgoal-conditioned policy

---
**Input:** subgoal-conditioned policy $\pi_\theta(\mathbf{a_t} \mid \mathbf{s_t}, \mathbf{g_t})$, subtask patterns $\mathbf{O}$, initial state $\tilde{s}_0$. ($\mathcal{O}_{-1}^{(j)}$ denotes the last state in subtask $j$ that has value and can has a smaller index than $L$.)
**Output:** Collected sequence $\{\tilde{\mathbf{s}}_t, \tilde{\mathbf{a}}_t, \tilde{\mathbf{g}}_t\}$.
**Procedure** Executing subgoal-conditioned policy $\pi_\theta$
  **while** task is not done **do**
    Query the most potential initial subgoal $\tilde{\mathbf{g}}_0 = \arg\min_j \|\tilde{\mathbf{s}}_0 - \mathcal{O}_0^{(j)}\|$.
    Set current subgoal $\tilde{\mathbf{g}} = \tilde{\mathbf{g}}_0$.
    **while** subtask has not terminated **do**
      Take action $\mathbf{a_t} = \arg\max_{\mathbf{a}} \pi_\theta(\tilde{\mathbf{a}} \mid \mathbf{s_t}, \tilde{\mathbf{g}})$
      Observe next state $s_{t+1}$
      Terminate if $\|\tilde{\mathbf{s}}_t - \mathcal{O}_{-1}^{(j)}\| \leq \epsilon$
    **end while**
    Query the most potential next subgoal $\tilde{\mathbf{g}} = \arg\min_j \|\tilde{\mathbf{s}}_t - \mathcal{O}_0^{(j)}\|$
  **end while**
**Return:** Collected sequence $\{\tilde{\mathbf{s}}_t, \tilde{\mathbf{a}}_t, \tilde{\mathbf{g}}_t\}$.

---

# D Experimental Details

## D.1 Data Generation for Synthetic Color Dataset

For $Color$-3, there are 3 types of patterns: 3 consecutive red, 3 consecutive yellow, and 3 consecutive blue. Each pattern is either generated independently (*Simple*) or correlated with the previous pattern (*Conditional*). For $Color$-10, there are 2 patterns in the dataset:{3 consecutive red + 3 consecutive yellow + 4 consecutive blue} and {3 consecutive blue + 3 consecutive yellow + 4 consecutive red}. Subgoals indicate the type of pattern to take. One sequence has a length of $T = 300$ for both $Color$-3 and $Color$-10, and we collected 100 sequences for each.

Because we want to conduct CI tests on the dataset, we need low-dimensional variables. Each state is represented by a variable with a mean of $1, 2, 3$ for red, yellow, and blue respectively, and an additive Gaussian noise. Each subgoal is a discrete variable indicating which color is selected for the next time step, taking one of the values of $\{0, 1, 2\}$, repeating the same value for 3 steps plus noise. Each action is the difference between the current state and the subgoal, i.e. $a_t = s_t - g_t$, then taking the action means that the next state is updated with $s_{t+1} = s_t + a_t + \epsilon$, where $\epsilon \sim \mathcal{N}(0, 0.01)$. In $Color$-3 (Simple), each pattern is selected independently, and in $Color$-3 (Conditional), we added one more color purple to it, represented by a mean value of $4$. Every pattern is also repeated for 3 steps. The conditional comes by re-coloring the yellow pattern generated in the $Color$-3 (Simple) dataset to purple if its precedent color is yellow or blue.

## D.2 Data Configuration in Kitchen

**Transferring to new tasks with $4$ subtasks**   We use kettle $\rightarrow$ bottom burner $\rightarrow$ top burner $\rightarrow$ slide cabinet and microwave $\rightarrow$ bottom burner $\rightarrow$ top burner $\rightarrow$ hinge cabinet in the demonstrations for training, and use two different tasks for testing: (a) kettle $\rightarrow$ bottom burner $\rightarrow$ top burner $\rightarrow$ hinge cabinet, and (b) microwave $\rightarrow$ bottom burner $\rightarrow$ top burner $\rightarrow$ slide cabinet. The difference in the target tasks' subtask composition is marked in blue.

**Transferring to new tasks with $5$-subtasks (generalized)**   To answer the question of whether the algorithm is able to generalize to a longer horizon, we test it on a target task of 5 subtasks to make it a more challenging scenario. Specifically, we use bottom burner $\rightarrow$ top burner $\rightarrow$ slide cabinet $\rightarrow$ hinge cabinet, and microwave $\rightarrow$ bottom burner $\rightarrow$ top burner $\rightarrow$ hinge cabinet in the demonstrations for training. For testing, we require the agent to manipulate: microwave $\rightarrow$ bottom burner $\rightarrow$ top burner $\rightarrow$ slide cabinet $\rightarrow$ hinge cabinet in order.

## D.3 CI Tests on $Color$ Dataset.

For Color Dataset, we calculate the mean of p-values across sequences, as is shown in Tab. 2. The answer for (1) whether $\mathbf{s_t} \not\perp\!\!\!\perp \mathbf{a_t} \mid \mathbf{g_t}$ , (2) whether $\mathbf{g_t} \not\perp\!\!\!\perp \mathbf{a}_{t+1} \mid \mathbf{g_t}$, and (3) $\mathbf{s}_{t+1} \perp\!\!\!\perp \mathbf{g_t} \mid \mathbf{s}_t, \mathbf{a}_t$ are all yes.

| CI test | $Color$-3 | $Color$-10 |
|---|---|---|
| (1) $\mathbf{s_t} \perp\!\!\!\perp \mathbf{a_t} \mid \mathbf{g_t}$ | 0.00 | 0.00 |
| (2) $\mathbf{g_t} \perp\!\!\!\perp \mathbf{a}_{t+1} \mid \mathbf{g}_{t+1}$ | 0.00 | 0.00 |
| (3) $\mathbf{s}_{t+1} \perp\!\!\!\perp \mathbf{g}_t \mid \mathbf{s}_t, \mathbf{a}_t$ | 0.99 | 0.99 |

Table 2: P-values for CI tests in $Color$-3 and -10.

## D.4 Supplementary Results for Seq-NMF

**Visualization of the learned matrices.**   The direct visualization of matrix $\mathbf{H}$ and $\mathbf{O}$ of seq-NMF on all three datasets are shown in Fig. 13. In each subfigure, matrix $\mathbf{H}$ is shown at the top, with each "row" of $\mathbf{H}$ representing a subgoal (0 or 1, indicating whether a subgoal is selected or not), and matrix $\mathbf{O}$ is shown on the left, with each "column" of $\mathbf{O}$ representing a subtask. Different subtasks are marked with different colors. The matrix in the middle is the convolutional product of $\mathbf{O}$ and $\mathbf{H}$, which is the trajectory matrix $\mathbf{X}$.

In Fig. 13 (a), we show the results on the $Color$-10 dataset. Two significant subtask patterns are shown on the left ({red, yellow, blue} and {blue, yellow, red}), each pattern lasts for 10 steps. On the top, $\mathbf{H}$ has spikes that indicate these patterns' corresponding appearance in the data matrix. Fig. 13 (b) shows the results on the Driving dataset and (c) shows the results on the Kitchen dataset, subtasks and subgoals are shown similarly.

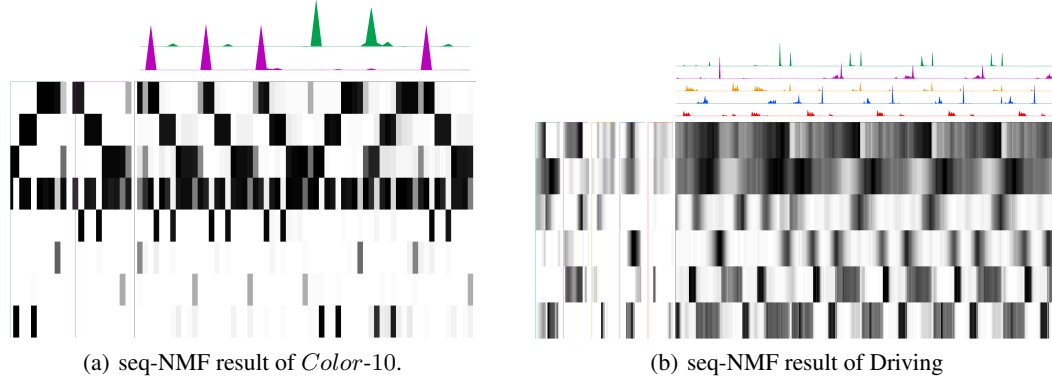

(a) seq-NMF result of $Color$-10.    (b) seq-NMF result of Driving

Figure 12: Results on the Kitchen dataset on new tasks.

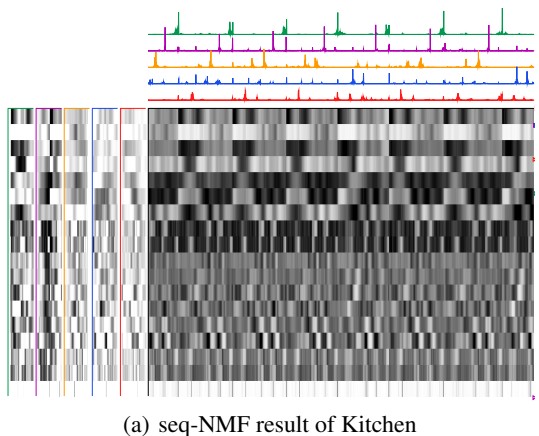

(a) seq-NMF result of Kitchen

Figure 13: Results on the Kitchen dataset on new tasks.

**Conversion from H to G**    We convert the learned binary indicator matrix **H** and the subtask patterns **O** to the subgoal **G** as follows:

$$g_{tj} = \frac{\sum_{\ell=1}^{L-1} O_{dj\ell} H_{j(t-\ell)}}{\sum_{j=1}^{J} \sum_{\ell=0}^{L-1} O_{dj\ell} H_{j(t-\ell)}}. \tag{6}$$

**Visualization of the learned subtask patterns.**    We use the above equation to turn the learned matrices into "options", and mark every time-bin with the corresponding option. The result of Driving is shown in the main paper in Fig. 6. As for the $Color$-10 data which consists of two patterns, we detect these subtask patterns co-evolving in the whole sequence, marked red and blue, respectively. Each subtask pattern last for 10 steps, which aligns with the true data generation process. When there are consecutive same subtask patterns, boundaries are not marked here but can be found in the learned matrix **H**.

**Numerical Evaluation.**    We compare the results of our method on the $Color$-3 (Simple) and $Color$-3 (Conditional) datasets to VTA (Kim et al. [2019]) and LOVE (Jiang et al.) in Tab. 3. We show the precision, recall, and F1 score of both methods in recognizing the correct boundaries of subtasks, which is the indicator terms in **H** of our method. Our method is not designed for high-dimensional data, so we constructed our own dataset, but for the result of LOVE, we directly take the results

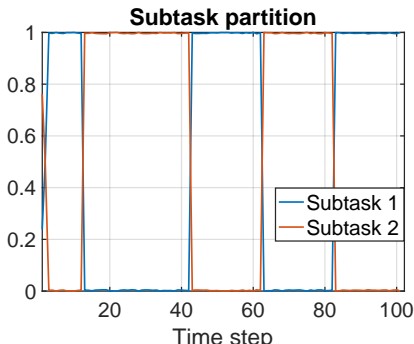

Figure 14: seq-NMF result on $Color$-10. The dominance of each subtask in explaining 10 sequences.

reported in Jiang et al.. Note that, the raw data of color frames
can be simply compressed to a single dimension by various
dimension reduction methods, after which we can use the compressed data to conduct our experiments.

| | $Color$-3 (Simple) | | | $Color$-3 (Conditional) | | |
|---|---|---|---|---|---|---|
| | **VTA** | **LOVE** | **Ours** | **VTA** | **LOVE** | **Ours** |
| Precision | $0.87 \pm 0.19$ | $0.99 \pm 0.01$ | $0.99 \pm 0.00$ | $0.84 \pm 0.22$ | $0.99 \pm 0.01$ | $0.99 \pm 0.00$ |
| Recall | $0.79 \pm 0.13$ | $0.85 \pm 0.03$ | $1.00 \pm 0.00$ | $0.82 \pm 0.16$ | $0.83 \pm 0.06$ | $1.00 \pm 0.00$ |
| F1 | $0.82 \pm 0.13$ | $0.91 \pm 0.02$ | $0.99 \pm 0.00$ | $0.83 \pm 0.19$ | $0.90 \pm 0.03$ | $0.99 \pm 0.00$ |

Table 3: Effect of our method on the $Color$-3 (Simple) and $Color$-3 (Conditional) datasets (5 seeds) in terms of the precision, recall and F1 score for recovering the correct boundaries of subtasks.

## D.5 Hyperparameters

For the experiments in seq-NMF, we use the following hyperparameters in Tab. 4. We use the same hyperparameters $\lambda_{sim}$, $\lambda_1$, and $\lambda_{bin}$ for all three datasets. Such hyperparameters are chosen by comparing the three penalty–when their values are approximately at the same level. Our results shows the hyperparameters chosen in this way are relatively robust such that one set of $\lambda_{sim}$, $\lambda_1$, and $\lambda_{bin}$ can fit all our three setting. As for the subtasks, we assume knowing the real number of subtasks for best performance, however we assume the algorithm also be able to detect subtask patterns as long as we have an upper bound estimation of the number of subtasks. The number of pattern duration $L$ is also the approximated upper bound for the max length of a subtask, and the table provide one reasonably good estimation.

| Hyperparameter | Value |
|---|---|
| $\lambda_{sim}$ | 0.0001 |
| $\lambda_1$ | 0.001 |
| $\lambda_{bin}$ | 0.01 |
| Number of subtasks (Color 3) | 3 |
| Number of subtasks (Color 10) | 2 |
| Number of subtasks (Driving) | 5 |
| Number of subtasks (Kitchen) | 5 |
| L (Color 3) | 3 |
| L (Color 10) | 10 |
| L (Driving) | 40 |
| L (Kitchen) | 100 |
| maxIter | 300 |
| start_bin_loss_iter | 30 |

Table 4: Hyperparameters in seq-NMF

For the experiments in transfering to new tasks, we use the following hyperparameters in Tab. 5. We use the same hyperparameters as Chen et al. [2023] for a fair comparison.

| Hyperparameter | Value |
|---|---|
| n_sample_per_epoch | 4096 |
| n_epoch | 2000 |
| *#Policy Config* | |
| activation | Relu |
| hidden_dim (policy) | (256, 256) |
| log_clamp_policy | (-20., 0.) |
| lr_policy | 3.e-4 |
| dim_c | 5 |
| *#PPO Config* | |
| hidden_dim (critic) | (256, 256) |
| lr_critic | 3.e-4 |
| use_gae | True |
| gamma | 0.99 |
| gae_tau | 0.95 |
| clip_eps | 0.2 |
| mini_batch_size | 1024 |
| *#IL Config* | |
| hidden_dim (discriminator) | (256, 256) |

Table 5: Hyperparameters in hierarchical IL

# E   Relation to the Probablistic Inference View of Reinforcement Learning

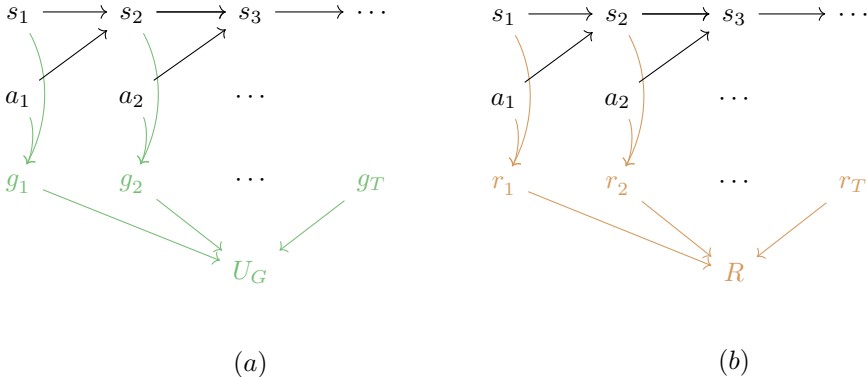

$(a)$ $(b)$

Figure 15: In Fig (a), we show the DAG built upon our subgoal framework, and in Fig (b), we add reward nodes as "optimality variables" that are always conditioned to be true for control under the framework by (Levine [2018]).

A standard task in reinforcement learning can be characterized by reward function $r(\mathbf{s}_t, \mathbf{a}_t)$. Solving an RL problem is searching for a policy that infers the most probable action sequence or most probable action distributions from states, such that:

$$\pi^\star = \arg \max_\pi \sum_{t=1}^{T} E_{(\mathbf{s}_t, \mathbf{a}_t) \sim \pi(\mathbf{a}_t | \mathbf{s}_t)} \left[ r(\mathbf{s}_t, \mathbf{a}_t) \right]. \tag{7}$$

This optimization procedure means that we seek a policy that maximizes the expected reward. The question is, how can we represent this problem in a probabilistic graphical model (PGM), such that estimating $\pi(\mathbf{a}_t \mid \mathbf{s}_t)$ from the PGM is equivalent to solving the optimization problem of maximizing the expected accumulated reward, as above?

Levine [2018] build the graphical model by laying out the states and actions that form the backbone, and add reward nodes as "optimality variables" that are conditioned to be true for an optimal policy, as is shown in Fig. 15. The optimality variable $z_t = 1$ denotes that the action is optimal, and $z_t = 0$ denotes that the action is not optimal. Then, we would like to show that when conditioning on the optimality variable $z_t$, the optimal action sequence is the one that maximizes the expected reward:

$$p(\tau \mid \mathbf{z}_{1:T}) \propto p(\tau, \mathbf{z}_{1:T}) = p(\mathbf{s}_1) \prod_{t=1}^{1} p(z_t = 1 \mid \mathbf{s}_t, \mathbf{a}_t) p(\mathbf{s}_{t+1} \mid \mathbf{s}_t, \mathbf{a}_t)$$

$$= p(\mathbf{s}_1) \prod_{t=1}^{T} \exp(r(\mathbf{s}_t, \mathbf{a}_t)) p(\mathbf{s}_{t+1} \mid \mathbf{s}_t, \mathbf{a}_t) \tag{8}$$

$$= \left[ p(\mathbf{s}_1) \prod_{t=1}^{T} p(\mathbf{s}_{t+1} \mid \mathbf{s}_t, \mathbf{a}_t) \right] \exp\left( \sum_{t=1}^{T} r(\mathbf{s}_t, \mathbf{a}_t) \right),$$

when we choose the conditional distribution over $z_t$ to be $p(z_t = 1 \mid \mathbf{s}_t, \mathbf{a}_t) = \exp(r(\mathbf{s}_t, \mathbf{a}_t))$. The first term on the right-hand side of Eqn. (8) is completely determined by the system's dynamics, which would be a constant if the system is deterministic. Then, the probability of observing a trajectory is in proportion to the exponential of the accumulated reward:

$$p(\tau \mid \mathbf{z}_{1:T}) \propto \mathbb{1}[p(\tau) \neq 0] \exp\left( \sum_{t=1}^{T} r(\mathbf{s}_t, \mathbf{a}_t) \right), \tag{9}$$

where indicator $\mathbb{1}[p(\tau) \neq 0]$ indicate that the trajectory is a feasible one. Then, maximizing the posterior to get a sequence of optimal actions in this graphical model is equivalent to maximizing the expected reward, which gives the distribution of $\pi(\mathbf{a}_t \mid \mathbf{s}_t, z_t = 1)$.

**Implication**   If we take the selection variable to be a scalar, then the posterior conditional policy we derive from the DAG in Fig. 15(a) gives $\pi(\mathbf{a}_t \mid \mathbf{s}_t, g_t = 1)$ (can be understood as achieving a single goal). Similar to the above analysis, such posterior gives us the optimal policy in a standard reinforcement learning framework.

So, we successfully build the analogy of the selection structure we identified in Sec. 3.1 to a probabilistic inference view of reinforcement learning. It strengthens the connection between the subgoal framework and the reinforcement learning task of maximizing the expected reward, and provides a new perspective to understand the subtask discovery problem.

# F    Subtask Ambiguity

There are two types of ambiguities that we care about here in subtask partition, as is shown in Fig. 16.

The Type A ambiguity (Fig. 16(a)) is that different subtask patterns share similar parts, as is marked with red. This would result in a high value of $\mathbf{O} \overset{\top}{*} \mathbf{X}$ for both $\mathbf{O}_2$ and $\mathbf{O}_3$ during the first two subtasks, because they are both highly correlated to the data matrix during that time period. Then multiplying it by $\mathbf{H}$ (with time shift within $L$) yields a high value of this overall regularization term.

The Type B ambiguity (Fig. 16(b)) appears when subtask patterns are more granularly partitioned than desire. The pattern of light blue to dark blue is exhibited as one consistent pattern across data, but could be recognized as different finer patterns as is shown in the red box. This contradict with our intuition that subtasks should be sparse enough, so we can suppress this ambiguity by adding the sparsity penalty $\mathcal{R}_1$.

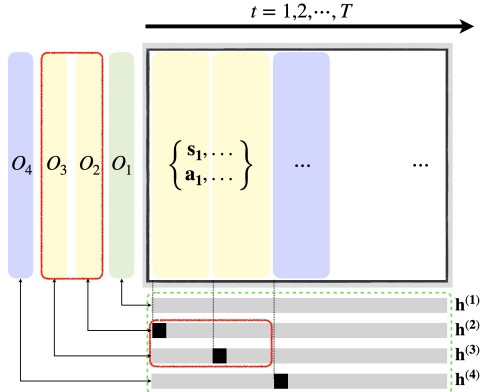
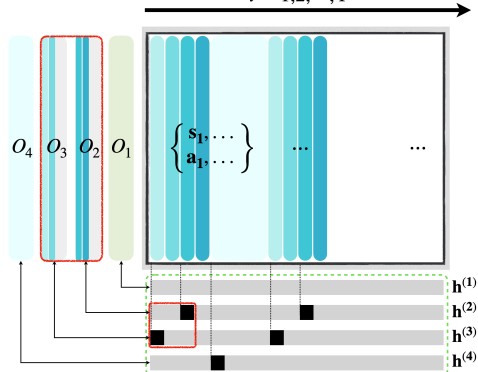

(a) Type A ambiguity: different subtask patterns share similar parts. Can be suppressed by $\mathcal{R}_{\text{sim}} = \|(\mathbf{O} \overset{\top}{*} \mathbf{X})\mathbf{S}\mathbf{H}^{\top}\|_{1, i \neq j}$.

(b) Type B ambiguity: subtask patterns are more granularly partitioned than desired. Can be suppressed by $\mathcal{R}_1 = \|\mathbf{H}\|_1$.

Figure 16: Different Types of ambiguities

# G    Derivation of Seq-NMF

At its core, NMF seeks to approximate a data matrix $\mathbf{V}$ as the product of two smaller non-negative matrices, $\mathbf{W}$ as base patterns or components and $\mathbf{H}$ as the coefficient matrix that describes how these patterns combine to approximate the original data. This can be expressed mathematically as:

$$\mathbf{V} \approx \mathbf{W} \times \mathbf{H}, \text{ where } \mathbf{W} = \begin{bmatrix} \mathbf{w}_1 & \mathbf{w}_2 & \cdots & \mathbf{w}_K \end{bmatrix}, \mathbf{H} = \begin{bmatrix} - & \mathbf{h}_1^T & - \\ - & \mathbf{h}_2^T & - \\ & \vdots & \\ - & \mathbf{h}_K^T & - \end{bmatrix} \tag{10}$$

## G.1    Multiplicative Update for Standard NMF

In standard NMF, where the optimization problem we try to solve is:

$$(\widetilde{\mathbf{W}}, \widetilde{\mathbf{H}}) = \arg \min_{\mathbf{W}, \mathbf{H}} \mathcal{L}(\mathbf{W}, \mathbf{H}),$$
$$\mathcal{L}(\mathbf{W}, \mathbf{H}) = \frac{1}{2}\|\widetilde{\mathbf{V}} - \mathbf{V}\|_F^2 \tag{11}$$
$$\widetilde{\mathbf{W}}, \widetilde{\mathbf{H}} \geq 0,$$

, the optimization problem is convex w.r.t $\mathbf{W}, \mathbf{H}$ separately, but not altogether. By alternatingly updating $\mathbf{W}$ and $\mathbf{H}$, we obtain the following gradient descent steps (additive update):

$$\frac{\partial}{\partial \mathbf{W}} \mathcal{L}(\mathbf{W}, \mathbf{H}) = \widetilde{\mathbf{X}} \mathbf{H}^\top - \mathbf{X} \mathbf{H}^\top$$
$$\frac{\partial}{\partial \mathbf{H}} \mathcal{L}(\mathbf{W}, \mathbf{H}) = \mathbf{W}^\top \widetilde{\mathbf{X}} - \mathbf{W}^\top \mathbf{X}. \tag{12}$$

Thus, gradient descent steps for $\mathbf{W}$ and $\mathbf{H}$ are:

$$\mathbf{W} \leftarrow \mathbf{W} - \eta_{\mathbf{W}} \left( \widetilde{\mathbf{X}} \mathbf{H}^\top - \mathbf{X} \mathbf{H}^\top \right)$$
$$\mathbf{H} \leftarrow \mathbf{H} - \eta_{\mathbf{H}} \left( \mathbf{W}^\top \widetilde{\mathbf{X}} - \mathbf{W}^\top \mathbf{X} \right),$$

with $\eta_{\mathbf{W}}$ and $\eta_{\mathbf{H}}$ being the learning rates. To avoid resulting in negative values, it is more desirable to use multiplicative updates. By setting:

$$\eta_{\mathrm{W}} = \frac{\mathbf{W}}{\mathbf{W} \mathbf{H} \mathbf{H}^\top}$$
$$\eta_{\mathrm{H}} = \frac{\mathbf{H}}{\mathbf{W}^\top \mathbf{W} \mathbf{H}},$$

the equivalent multiplicative updates becomes (Lee and Seung [1999]):

$$\mathbf{W} \leftarrow \mathbf{W} \times \frac{\mathbf{X} \mathbf{H}^\top}{\mathbf{W} \mathbf{H}^\top} = \mathbf{W} \times \frac{\mathbf{X} \mathbf{H}^\top}{\widetilde{\mathbf{X}} \mathbf{H}^\top}$$
$$\mathbf{H} \leftarrow \mathbf{H} \times \frac{\mathbf{W}^\top \mathbf{X}}{\mathbf{W}^\top \mathbf{W} \mathbf{H}} = \mathbf{H} \times \frac{\mathbf{W}^\top \mathbf{X}}{\mathbf{W}^\top \widetilde{\mathbf{X}}}. \tag{13}$$

## G.2 Multiplicative update for seq-NMF

Then, we incorporate the time element. Applying the above derivation for standard NMF for each time-bin delay $\ell$, we get the following update rules for convNMF (Smaragdis [2004]):

$$\mathbf{W}_{..\ell} \leftarrow \mathbf{W}_{..\ell} \times \frac{\mathbf{X} \overset{\ell \rightarrow}{\mathbf{H}^\top}}{\widetilde{\mathbf{X}} \overset{\ell \rightarrow}{\mathbf{H}^\top}}$$

$$\mathbf{H} \leftarrow \mathbf{H} \times \frac{\sum_\ell \mathbf{W}_{..\ell}^\top \overset{\leftarrow \ell}{\mathbf{X}}}{\sum_\ell \mathbf{W}_{..\ell}^\top \overset{\leftarrow \ell}{\widetilde{\mathbf{X}}}} = \mathbf{H} \times \frac{\mathbf{W} \overset{\leftarrow}{*} \mathbf{X}}{\mathbf{W} \overset{\leftarrow}{*} \widetilde{\mathbf{X}}}$$

Where the operator $\ell \rightarrow$ shifts a matrix in the $\rightarrow$ direction by $\ell$ timebins, which implies a delay by $\ell$ timebins, and $\leftarrow \ell$ shifts a matrix in the $\leftarrow$ direction by $\ell$ timebins. When $L = 1$, convNMF is reduced to standard NMF.

## G.3 Derivation of Regularizer Terms in Standard NMF

Adding the regularizer terms in standard NMF, we get the following additive update rules similar to Eqn. (12):

$$\frac{\partial \mathcal{L}}{\partial \mathbf{W}} = \widetilde{\mathbf{X}} \mathbf{H}^\top - \mathbf{X} \mathbf{H}^\top + \frac{\partial \mathcal{R}}{\partial \mathbf{W}}$$
$$\frac{\partial \mathcal{L}}{\partial \mathbf{H}} = \mathbf{W}^\top \widetilde{\mathbf{X}} - \mathbf{W}^\top \mathbf{X} + \frac{\partial \mathcal{R}}{\partial \mathbf{H}}.$$

We set:

$$\eta_{\mathbf{W}} = \frac{\mathbf{W}}{\widetilde{\mathbf{X}} \mathbf{H}^\top + \frac{\partial \mathcal{R}}{\partial \mathbf{W}}}$$
$$\eta_{\mathbf{H}} = \frac{\mathbf{H}}{\mathbf{W}^\top \widetilde{\mathbf{X}} + \frac{\partial \mathcal{R}}{\partial \mathbf{H}}},$$

and turns it into multiplicative update:

$$\mathbf{W} \leftarrow \mathbf{W} - \eta_{\mathbf{W}} \frac{\partial \mathcal{L}}{\partial \mathbf{W}} = \mathbf{W} \times \frac{\mathbf{X} \mathbf{H}^\top}{\widetilde{\mathbf{X}} \mathbf{H}^\top + \frac{\partial \mathcal{R}}{\partial \mathbf{W}}}$$
$$\mathbf{H} \leftarrow \mathbf{H} - \eta_{\mathbf{H}} \frac{\partial \mathcal{L}}{\partial \mathbf{H}} = \mathbf{H} \times \frac{\mathbf{W}^\top \mathbf{X}}{\mathbf{W}^\top \widetilde{\mathbf{X}} + \frac{\partial \mathcal{R}}{\partial \mathbf{H}}}. \tag{14}$$

### G.4 Derivation of Regularizer Terms in Seq-NMF (Ours)

Applying Eqn. (14) at each time-bin, we get the following multiplicative update rules for convNMF:

$$
\mathbf{W}_{..\ell} \leftarrow \mathbf{W}_{..\ell} \times \frac{\mathbf{X}\,\overset{\ell\rightarrow}{\mathbf{H}}^{\top}}{\widetilde{\mathbf{X}}\,\overset{\ell\rightarrow}{\mathbf{H}}^{\top} + \frac{\partial \mathcal{R}}{\partial \mathbf{W}_{..\ell}}}
$$

$$
\mathbf{H} \leftarrow \mathbf{H} \times \frac{\mathbf{W}\,\overset{\leftarrow}{*}\,\mathbf{X}}{\mathbf{W} * \widetilde{\mathbf{X}} + \frac{\partial \mathcal{R}}{\partial \mathbf{H}}}
\tag{15}
$$

For each regularizer $\mathcal{R}_{\mathrm{bin}}$, $\mathcal{R}_1$ and $\mathcal{R}_{\mathrm{sim}}$ in our formulation in Sec. 3.3, we separately derive the corresponding gradient used for multiplicative update (we change the matrix $\mathbf{W}$ to $\mathbf{O}$, keeping it aligned with the main part of this paper).

$$
\begin{cases}
\dfrac{\partial \mathcal{R}_{\mathrm{bin}}}{\partial \mathbf{H}} = \lambda_{\mathrm{bin}}\left(\mathbf{H} \odot \mathbf{T}_0 \odot \mathbf{T}_0 - \mathbf{H} \odot \mathbf{H} \odot \mathbf{T}_0\right),\,(\mathbf{T}_0 = \mathbf{1} - \mathbf{H}) \\[2mm]
\dfrac{\partial \mathcal{R}_1}{\partial \mathbf{H}} = \lambda_1(\mathbf{1} - \mathbf{I})\mathbf{H} \\[2mm]
\dfrac{\partial \mathcal{R}_{\mathrm{sim}}}{\partial \mathbf{H}} = \lambda_{\mathrm{sim}}(\mathbf{1} - \mathbf{I})\mathbf{O}\,\overset{\leftarrow}{*}\,\mathbf{X}\mathbf{S}
\end{cases}
\tag{16}
$$

$$
\begin{cases}
\dfrac{\partial \mathcal{R}_{\mathrm{bin}}}{\partial \mathbf{O}} = \mathbf{0} \\[2mm]
\dfrac{\partial \mathcal{R}_1}{\partial \mathbf{O}} = \mathbf{0} \\[2mm]
\dfrac{\partial \mathcal{R}_{\mathrm{sim}}}{\partial \mathbf{O}_{..\ell}} = \lambda_{\mathrm{sim}}\,\overset{\leftarrow\ell}{\mathbf{X}}\,\mathbf{S}\mathbf{H}^{\top}(\mathbf{1} - \mathbf{I})
\end{cases}
\tag{17}
$$

Therefore, the overall updating rules are:

$$
\mathbf{O}_{..\ell} \leftarrow \mathbf{O}_{..\ell} \times \frac{\mathbf{X}\,\overset{\ell\rightarrow}{\mathbf{H}}^{\top}}{\widetilde{\mathbf{X}}\,\overset{\ell\rightarrow}{\mathbf{H}}^{\top} + \lambda_{\mathrm{sim}}\,\overset{\leftarrow\ell}{\mathbf{X}}\,\mathbf{S}\mathbf{H}^{\top}(\mathbf{1} - \mathbf{I})}
$$

$$
\mathbf{H} \leftarrow \mathbf{H} \times \frac{\mathbf{O}\,\overset{\leftarrow}{*}\,\mathbf{X}}{\mathbf{O} * \widetilde{\mathbf{X}} + \lambda_{\mathrm{bin}}\left(\mathbf{H} \odot \mathbf{T}_0 \odot \mathbf{T}_0 - \mathbf{H} \odot \mathbf{H} \odot \mathbf{T}_0\right) + \lambda_1(\mathbf{1} - \mathbf{I})\mathbf{H} + \lambda_{\mathrm{sim}}(\mathbf{1} - \mathbf{I})\mathbf{O}\,\overset{\leftarrow}{*}\,\mathbf{X}\mathbf{S}}
\tag{18}
$$

## H  Hardware

All experiments were conducted on either NVIDIA L40, or GeForce RTX 3080 Ti, or a Mac M1 chip with 16GB of RAM.

