# OpenReview forum: "Identifying Selections for Unsupervised Subtask Discovery"
_NeurIPS.cc/2024/Conference — NeurIPS 2024 poster_

### Official Review · Reviewer_1Ld6 · 2024-07-04

**Soundness:** 2
**Presentation:** 3
**Contribution:** 3
**Rating:** 6
**Confidence:** 3

**Summary:**

This paper addresses offline subtask discovery from a causal perspective by identifying subgoals as selections, targeting at solving long-horizon tasks and acquiring transferrable skills. The algorithm design is well-motivated and shows superior performance in offline subtask discovery.

**Strengths:**

(a) The causal-graph-based algorithm design is well motivated and theoretically solid.

(b) The presentation is elegant and easy to follow.

(c) Superiority of the proposed algorithm is shown through both quantitative and qualitative study.

**Weaknesses:**

(a) Kitchen is a challenging benchmark, but each of its tasks consists of a sequence of subtasks. Consequently, subtask discovery from this well-structured offline data is relatively straightforward. It would be beneficial to demonstrate the subtask discovery capability of the proposed algorithm in tasks that lack clearly defined, semantically obvious subtask structures.

(b) This work also focuses on multi-task learning, but it is confined to scenarios where the tasks in the set are merely different compositions of the same set of subtasks. More generalized multi-task learning can be future directions.

**Questions:**

Please see the weakness part.

---

> ### Author Rebuttal · Authors · 2024-08-06
>
> We are profoundly thankful for your valuable feedback and the time you have dedicated to reading it, as they will surely improve the quality of this manuscript. In light of your suggestion, we incorporated additional discussions, as well as experiments to demonstrate the generalizability of our method.
>
> **Q1**: *“Kitchen is a challenging benchmark, but each of its tasks consists of a sequence of subtasks. Consequently, subtask discovery from this well-structured offline data is relatively straightforward. It would be beneficial to demonstrate the subtask discovery capability of the proposed algorithm in tasks that lack clearly defined, semantically obvious subtask structures.”*
>
> **A1**: We completely agree with you that, as you mentioned, Kitchen is a well-structured dataset thus it is easier to consider. Further, we aim to take it as a starting point to develop a principled way to tackle subtask discovery problems. More practical issues will surely need to be discussed in the future.
>
> In our setting, we deal with multi-task learning problems when new tasks are arbitrarily composed by learned subtasks. Like others, we do not use the skill structure annotations in either training or testing, so the semantically meaningful "subtasks" used in the paper are simply a way to refer to the learned segmentations. It is not necessary to have them pre-defined but it is good to have them well-structured for easier evaluation.
>
> At the same time, we do not rely on a more restricted subtask structure than other work. we adopt a common setting in the literature regarding the subtask discovery problem. By conducting a comprehensive literature review on all the datasets that subtask-discovery work considers, we found that in order to evaluate the methods that are proposed and also for better understanding, it is common that we use tasks that are naturally composed of semantically meaningful and clear subtasks.
>
> For example, the latest works: [1] consider a grid-world setting BabyAI [2] that uses well-structured language (e.g.GoTo, Pickup) to indicate subgoals to accomplish, and ALFRED [3] environment that asks the agent to perform household tasks, both involve clearly defined subtasks like navigating in the kitchen, putting into the microwave, etc. Another work [4] considers MetaWorld, training on tasks that are composed of a sequence of subtasks like grasp, and move.
>
> Overall, we do agree with your insight that the semantically clear-defined subtask structure is a straightforward scenario. Meanwhile, this kind of subtask structure is also employed by all related works, and we think that the Kitchen environment is a reasonable choice for this first step of exploring a theoretical framework for subtask-discovery, and your great suggestion inspires us to see how the algorithm would perform in other environments.
>
> [1] Fu, Haotian, et al. "Language-guided Skill Learning with Temporal Variational Inference." In Forty-first International Conference on Machine Learning (ICML 2024).
>
> [2] Chevalier-Boisvert, et al. Babyai: A platform to study the sample efficiency of grounded language learning. In 7th International Conference on Learning Representations, ICLR 2019, New Orleans, LA, USA, May 6-9, 2019. OpenReview.net, 2019.
>
> [3] Shridhar, M., et al. ALFRED: A benchmark for interpreting grounded instructions for everyday tasks. In 2020 IEEE/CVF Conference on Computer Vision and Pattern Recognition, CVPR 2020, Seattle, WA, USA, June 13-19, 2020, pp. 10737–10746. Computer Vision Foundation / IEEE, 2020a.
>
> [4] Zheng, Ruijie, et al"PRISE: LLM-Style Sequence Compression for Learning Temporal Action Abstractions in Control." In Forty-first International Conference on Machine Learning (ICML 2024).
>
>
> **Q2**: *“This work also focuses on multi-task learning, but it is confined to scenarios where the tasks in the set are merely different compositions of the same set of subtasks. More generalized multi-task learning can be a future direction.”*
>
> **A2**: We sincerely appreciate you bringing this to our attention. Generalization to new scenarios that encompass unseen subtasks, i.e. when no discovered subtasks are reusable, can be challenging and deserves further investigation.
>
> In light of your great suggestion, we have made the following efforts, including new experiments in our revision:
>
> * *Possible solutions*: We provide some preliminary solutions to such a problem that you have insightfully pointed out in more generalized scenarios: for example, considering the ability to discover new subtasks when the target subtask is out-of-distribution of the training set; learning subtasks more granularly such that more primitive actions can serve to facilitate new tasks learning.
>
> * *Additional experiments*: We conducted additional experiments under in Kitchen, and tested the method’s generalizability to a new task composed of one more subtask, i.e. $5$ sequential manipulation subtasks. Such generalization to longer-horizon tasks is challenging and is not taken into account by our MIL baseline[3]. Please refer to the uploaded pdf for the result and the **Global Response** for experimental details.
>
> We deal with scenarios where all training and testing tasks share the same set of subtasks, while with different (unseen) compositions, just like other related works [1] [2] [3], and have validated our method through challenging experiments. We are greatly thankful for your suggestion, as it undoubtedly encourages us to pursue further investigation.
>
> [1] Yifeng Zhu, et al. Bottom-up skill discovery from unsegmented demonstrations for long-horizon robot manipulation. IEEE Robotics and Automation Letters, 7(2):4126–4133, 2022.
>
> [2] Yiding Jiang, et al. Learning Options via Compression.
>
> [3] Jiayu Chen, et al. Multi-task Hierarchical Adversarial Inverse Reinforcement Learning. In Proceedings of the 40th International Conference on Machine Learning, pages 4895–4920. PMLR, July 2023.

---

> > ### Comment · Reviewer_1Ld6 · 2024-08-12
> >
> > Thanks for the detailed feedback. I would keep my rating since not many revisions have been made.

---

### Official Review · Reviewer_imJV · 2024-07-12

**Soundness:** 3
**Presentation:** 4
**Contribution:** 3
**Rating:** 8
**Confidence:** 2

**Summary:**

The paper studies the subtask decomposing problem. The paper proposes a formal definition of subtasks as the outcome of selections. The proposed seq-NMF is introduced to learn the subgoals and extract subtasks, conforming with the proposed theory. The experimental results show strong results on transferring to new tasks.

**Strengths:**

- The paper is well-written and easy to follow.
- The paper offers a formal definition of subtasks, to address the challenge of interpretability that has not been discussed in the prior works.
- The insight of the definition of segmented subtasks should be consistent with the true generating process is novel and interesting.
- The paper validates the proposed theory through the unseen long-horizon manipulation tasks and the results present a strong performance of the proposed method.

**Weaknesses:**

- The paper only considers state sequences to extract the subgoals and subtasks. In real-world settings, the states are not available. So it is not clear if the proposed theory is also applicable to the real world with various modalities, e.g., visual perception. It would be nice to explore and discuss the possibility.
- The generalization of the proposed definition of the subtask is rather specific and the shift of task distribution is not significant. So the generalization of the proposed method might be less convincing.

**Questions:**

See weaknesses.

**Limitations:**

Yes, the limitations are discussed in the paper.

---

> ### Author Rebuttal · Authors · 2024-08-06
>
> We appreciate your thorough review and the constructive insights you provided, which will undoubtedly enhance the quality of our manuscript. In response to your comments, we have included several new discussions in the revised manuscript, and additional experiments, particularly addressing more generalized multi-task learning scenarios.
>
>
>
>
> **Q1**: *“The paper only considers state sequences to extract the subgoals and subtasks. In real-world settings, the states are not available. So it is not clear if the proposed theory is also applicable to the real world with various modalities, e.g., visual perception. It would be nice to explore and discuss the possibility.”*
>
>
> **A1**: The point that you raised is insightful and inspiring, and application to various data modalities will surely be an important aspect. It requires different techniques, and there are certainly a lot of practical issues to consider. In this work, we mainly focus on a theoretical analysis of the understanding of subtasks, and on rigorously validating our interpretation.
>
>
> * *Possible solutions*: When states are not available, it is still possible to extract useful representation from raw input data. For example, the CI test for high-dimensional data can be performed on the low-dimensional embedding level, followed by matrix decomposition and hierarchical imitation learning. In order to obtain the embeddings that are beneficial for downstream tasks, we essentially turn to a representation learning problem that learns a compact way to encode the sufficient feature for subtask identification and transfer, while mitigating the effect of redundant features. Various techniques can be applied to facilitate this representation extraction process [1] [2]. It is intuitively applicable that the proposed method adapts to more data modalities. We believe that our results will provide inspiration to the community for exploring more general scenarios, including multi-modalities data. It is a direction that is worth further investigation.
>
>
>
> [1] Huang, Biwei, et al. "Adarl: What, where, and how to adapt in transfer reinforcement learning." arXiv preprint arXiv:2107.02729 (2021).
>
>
> [2] Liu, Yuren, et al. "Learning world models with identifiable factorization." Advances in Neural Information Processing Systems 36 (2023): 31831-31864.
>
>
> **Q2**: *“The generalization of the proposed definition of the subtask is rather specific and the shift of task distribution is not significant. So the generalization of the proposed method might be less convincing.”*
>
> **A2**: We sincerely thank you for pointing out this possible limitation. In order for a wider generalization, we need to consider more practical issues. Your insight inspires us to provide a more general answer to the problem such that we are able to handle more distribution shift scenarios.
>
>
> In light of your suggestion, we have made the following efforts, including new experiments, in the updated manuscript:
>
>
> * *More experimental results on distribution shift*: We conducted additional experiments by considering a distribution shift problem that involves longer-horizon tasks, and incorporated the results in the uploaded pdf. Specifically, under the Kitchen environment, we keep the same training set of tasks (each task is composed of $4$ sequential manipulation subtasks), and test the method’s generalizability to a new task composed of one more subtask, i.e. $5$ random sequential manipulation subtasks. Such generalization to longer-horizon tasks is not taken into consideration by some of the existing works [3], and our empirical results show that our formulations are able to deal with such a more challenging distribution shift problem. Please refer to the **Global Response** for experiment details.
>
>
> * *Rationale for adopting the current generalization cases*: We totally agree with you that our settings do not encompass all kinds of task distribution shifts, but focus on the shift where new tasks are randomly composed of seen subtasks.
> This assumption on task distribution is also widely used in other subtask/skill discovery literature. For instance,  BUDS [1] inquires about its ability to solve new task variants that require different subtask combinations; LOVE [2] tests on the grid-world environment where each subtask corresponds to picking up a specific object; MH-AILR [3] also uses the Kitchen environment, in which the tasks require the sequential completion of four specific subtasks. In short, all works consider the generalization in cases where the training set of subtasks provides full support of the target set of subtasks; our generalization experiment in the Kitchen environment is not more limited than other works.
>
> * *Possible solutions*: We provide preliminary solutions to such a problem in more types of task distribution shifts: for example, utilizing RL combined with IL when the target subtask is out-of-distribution of the training set; learning subtasks on a more granular level such that more primitive actions can serve to facilitate the discovery of new subtasks. We believe that our results could serve as a theoretical basis, which may prove helpful for many future explorations by the community.
>
> Meanwhile, we take this work as a starting point, and we aim to develop a principled, easily-extensible way to resolve the subtask-discovery problem. We thank you for your great advice, which definitely inspired us to do so.
>
>
> [1] Yifeng Zhu, et al. Bottom-up skill discovery from unsegmented demonstrations for long-horizon robot manipulation. IEEE Robotics and Automation Letters, 7(2):4126–4133, 2022.
>
>
> [2] Yiding Jiang, Evan Zheran Liu, Benjamin Eysenbach, J Zico Kolter, and Chelsea Finn. Learning Options via Compression.
>
>
> [3] Jiayu Chen, Dipesh Tamboli, Tian Lan, and Vaneet Aggarwal. Multi-task Hierarchical Adversarial Inverse Reinforcement Learning. In Proceedings of the 40th International Conference on Machine Learning, pages 4895–4920. PMLR, July 2023.

---

> > ### Comment · Reviewer_imJV · 2024-08-11
> > **Response for the authors**
> >
> > I appreciate the authors for their efforts in additional experiments and explanations that address my concern.
> > - The authors clarify the gap from real-world setup and provide a potential solution. I am convinced that the paper can be a first step in applying its theoretical framework for subtask discovery and further inspire work toward real-world subtask discovery.
> > - The authors strengthen the verification of generalization by testing the longer horizon of tasks.
> >
> > After reading all the reviews, which I believe have been well addressed by the response from the authors, I am willing to raise my score to strong accept.

---

> > > ### Author Response · Authors · 2024-08-12
> > > **Thank you for your feedback**
> > >
> > > Thank you for your encouragement, and we are also excited to explore more complex scenarios building upon the selection framework. We are glad that the additional clarifications and experiments are helpful. We thank you for the valuable insights you have contributed.

---

### Official Review · Reviewer_WHCR · 2024-07-12

**Soundness:** 3
**Presentation:** 3
**Contribution:** 3
**Rating:** 8
**Confidence:** 3

**Summary:**

This paper studies the problem of decomposing expert trajectories (in the context of imitation learning) into sub-trajectories corresponding to subtasks. First, the authors introduce a causal framework to understand and explain what subtasks mean in this context. Subtasks are then defined to be variables that reduce uncertainty in the observed expert actions (or selections in the language of causal analysis). Motivated by this definition, a matrix factorization based task decomposition algorithm is presented. Experiments on multiple environments  demonstrate that the algorithm is effective at discovering subtasks from expert trajectories.

**Strengths:**

- There are many novel aspects to the paper. The authors provide an argument to view subtasks as selection variables and this insight is used to develop the task decomposition algorithm presented in the paper. The task decomposition algorithm also seems novel and the use of matrix factorization here is very apt.
- Experiments presented in the paper indicate that the proposed approach is effective at discovering useful subtasks. The experiments also show that the subtask decomposition algorithm enables learning policies (in the context of imitation learning) that generalize well to new and unseen tasks. In this context of transfer learning, the proposed approach outperforms many existing state-of-the-art methods.
- I believe that the paper offers interesting insights relevant to the NeurIPS community and RL researchers working on hierarchical and compositional RL. These insights have the potential to inspire new directions of research.

**Weaknesses:**

One primary weakness is that some parts of the paper lack clarity. For instance, the mathematical objective in Equation (1) is not very clear  although the overall idea and intuitive definition is clear from the text. Defining the notations early on in the paper would make it much easier to read and understand.

**Questions:**

- In the second line of Equation (1), should the "forall" and "exists" terms be switched? Should it read as "for all sub-trajectories, there is exists a subtask..." ?
- The use of convolution in the matrix factorization algorithm suggests that it allows multiple subtasks to be "active" at a given step. Could you provide some intuition behind this?

**Limitations:**

I don't think there are any major limitations that need to be discussed in the paper.

---

> ### Author Rebuttal · Authors · 2024-08-06
>
> We are truly grateful for the time you have taken to review our work and for your insightful comments. Your valuable feedback has significantly enhanced the clarity of our manuscript.
>
> **Q1**: *“One primary weakness is that some parts of the paper lack clarity. For instance, the mathematical objective in Equation (1) is not very clear although the overall idea and intuitive definition is clear from the text. Defining the notations early on in the paper would make it much easier to read and understand.”*
>
> **A1**: We sincerely thank you for your great feedback and suggestions. Following your suggestion, we have carefully gone through the paper to make sure the notations are friendly. Specifically, in the updated version of the manuscript, we have defined the notations early before using them:
> > “**Problem Formulation and Notations**: Given the above context, we formulate the considered imitation learning problem as follows: we have a distribution of tasks $\mathcal{P}_e(\mathcal{T})$ and a corresponding set of expert trajectories $\mathcal{D}=\\{\tau_n\\}\_{n=1}^N$, and we aim to let the imitater learn a policy that can be transferred to new tasks that follow a different distribution $\mathcal{P}_i(\mathcal{T})$. Each task sampled from $\mathcal{P}\_{\cdot}(\mathcal{T})$ should be generated by a MDP and composed of a sequence of option $\{\mathcal{O}_j, \cdots\}$, where $\mathcal{O}_j =\langle \mathcal{I}_j, \pi_j, \beta_j \rangle _j$. We use $\mathbb{\mathcal{O}}=\bigcup\_{j=1}^J \mathcal{O}_j$ to denote all J options, and ${\xi\_{p}=\\{\mathbf{s_t},\mathbf{a_t}, ...\\}\_{t=1}^{\leq L} }$ as a sub-sequence of states and actions $(\mathbf{s_t}, \mathbf{a_t})$ from any trajectory $\tau_n$. Each trajectory can be partitioned into sub-sequences ${\xi\_{p}}$ with maximum length L.”
>
> **Q2**: *“In the second line of Equation (1), should the "forall" and "exists" terms be switched? Should it read as "for all sub-trajectories, there exists a subtask..."?”*
>
>
> **A2**: We highly appreciate your insightful question. We agree with your interpretation of "for all sub-trajectory, there exists a subtask such that the sub-trajectory is sampled from the option, ..." and adjust the "for all" and "exist" notation accordingly. You are completely correct and it is better to present the notations in the way you suggested.
>
> **Q3**: *“The use of convolution in the matrix factorization algorithm suggests that it allows multiple subtasks to be "active" at a given step. Could you provide some intuition behind this?”*
>
> **A3**: We sincerely thank you for the great question. We fully agree with you that, in practice, it is possible to have undesired duplicate activations (which could be due to optimization errors like local optimal solutions, etc.), while the intuition is that subtasks should have distinct patterns and be distinctively activated.
>
> At the same time, it might be worth noting that our formulation has already discouraged such situations. More specifically, given that our main insight is to view subgoals as binary selections, we enforce the learned coefficient matrix also to be $0/1$. Then if multiple subtasks are activated at the same time, the sparsity constraint $R_1$ as well as the redundancy constraint $R_\text{sim}$ would penalize those competing factors. Therefore, the goal of our regularization is to forbid the cases that you kindly mentioned, of which the effect has been empirically verified in our experiments and we observed no duplicated activations.

---

> > ### Comment · Reviewer_WHCR · 2024-08-12
> >
> > I thank the authors for the clarifications and the additional experiments. I think the paper is definitely clearer to read after the revision. All my questions have been addressed by the authors and I remain in support of accepting the paper and have also raised my score to reflect this.

---

> > > ### Author Response · Authors · 2024-08-12
> > > **Thank you for your feedback**
> > >
> > > We sincerely appreciate your detailed review and insightful suggestions. Thank you for checking the response and for your support.

---

### Author Rebuttal · Authors · 2024-08-06

We sincerely thank all the reviewers for your dedicated time and insightful comments. We are happy to see that the novelty of this work is well-recognized by all reviewers, which lies in the idea of identifying subtasks as selections, as well as the soundness of the corresponding algorithms. We added one additional experiment to test the generalization ability of the method to long-horizon tasks, details of which are provided below, and the result is presented in the figure of the uploaded pdf.

**Additional experiment in Kitchen**:
In Sec. 5.3, we have tested the performance of the proposed algorithm on a new task in the Kitchen environment [1] that is composed of $4$ subtasks with an unforeseen combination, while trained on expert demonstrations [2] with matching horizon ($4$ subtasks). A further question is, is the algorithm able to generalize to more scenarios?

In order to answer this question, we test on a target task of $5$ subtasks to make it a more challenging generalization scenario. Specifically, we use |bottom burner| $\rightarrow$ |top burner| $\rightarrow$ |slide cabinet| $\rightarrow$  |hinge cabinet| , and |microwave| $\rightarrow$ |bottom| |burner| $\rightarrow$ |top burner| $\rightarrow$ |hinge cabinet| in the demonstrations for training. For testing, we require the agent to manipulate: |microwave| $\rightarrow$ |bottom burner| $\rightarrow$ |top burner| $\rightarrow$ |slide cabinet| $\rightarrow$ |hinge cabinet| in order. We show the result (accumulated return w.r.t. training steps) in the additional figure.
The baseline method H-AIRL[3] has only been evaluated on a 4-subtask horizon, and our algorithm is demonstrated to have better performance compared with all baselines. This suggests that the subtasks that we learned from the demonstrations as selections show better generalization ability even in accomplishing challenging long-horizon tasks, further validating the superiority of our method.

[1] Justin Fu, Aviral Kumar, Ofir Nachum, George Tucker, and Sergey Levine.D4rl: Datasets for deep data-driven reinforcement learning. arXiv preprint arXiv:2004.07219, 2020.

[2] Abhishek Gupta, Vikash Kumar, Corey Lynch, Sergey Levine, and Karol Hausman. Relay Policy Learning: Solving Long-Horizon Tasks via Imitation and Reinforcement Learning, October 2019.

[3] Jiayu Chen, Dipesh Tamboli, Tian Lan, and Vaneet Aggarwal. Multi-task Hierarchical Adversarial Inverse Reinforcement Learning. In Proceedings of the 40th International Conference on Machine Learning, pages 4895–4920. PMLR, July 2023

---

### Decision · Program_Chairs · 2024-09-25

**Decision:**

Accept (poster)

**Comment:**

This paper introduces a new definition of subgoal as a selection variable in causal inference and proposes a sequential non-negative matrix factorization method to discover subgoals. The empirical result on the Kitchen environment shows that the proposed method can learn useful subtasks from expert demonstrations and outperform relevant baselines as a result.

All of the reviewers found that the proposed definition of subgoal is well-motivated and convincing, and the proposed method based on this definition is also technically sound. Although there were minor concerns around the tasks in the Kitchen environment not challenging enough, all of the reviewers unanimously agreed that the paper is novel and strong enough to be presented at NeurIPS. Thus, I recommend to accept this paper.